# Doublecortin and JIP3 are neural-specific counteracting regulators of dynein-mediated retrograde trafficking

Xiaoqin Fu[1,2,3]*[†], Lu Rao[4][†], Peijun Li[1,2,3], Xinglei Liu[4], Qi Wang[1], Alexander I Son[5], Arne Gennerich[4]*, Judy Shih-Hwa Liu[6]*

[1]The Second Affiliated Hospital and Yuying Children's Hospital, Wenzhou Medical University, Wenzhou, China; [2]Key Laboratory of Structural Malformations in Children of Zhejiang Province, Wenzhou, China; [3]Key Laboratory of Perinatal Medicine of Wenzhou, Wenzhou, China; [4]Department of Biochemistry and Gruss-Lipper Biophotonics Center, Albert Einstein College of Medicine, Bronx, United States; [5]Center for Neuroscience Research, Children's National Research Institute, Children's National Hospital, Washington, United States; [6]Department of Neurology, Department of Molecular Biology, Cell Biology, and Biochemistry, Brown University, Providence, United States

*For correspondence:
fuxq@wzhealth.com (XF);
arne.gennerich@einsteinmed.
edu (AG);
judy_liu@brown.edu (JS-HL)

[†]Co-first author

**Competing interest:** The authors declare that no competing interests exist.

**Abstract** Mutations in the microtubule (MT)-binding protein doublecortin (DCX) or in the MT-based molecular motor dynein result in lissencephaly. However, a functional link between DCX and dynein has not been defined. Here, we demonstrate that DCX negatively regulates dynein-mediated retrograde transport in neurons from $Dcx^{-/y}$ or $Dcx^{-/y};Dclk1^{-/-}$ mice by reducing dynein's association with MTs and disrupting the composition of the dynein motor complex. Previous work showed an increased binding of the adaptor protein C-Jun-amino-terminal kinase-interacting protein 3 (JIP3) to dynein in the absence of DCX. Using purified components, we demonstrate that JIP3 forms an active motor complex with dynein and its cofactor dynactin with two dyneins per complex. DCX competes with the binding of the second dynein, resulting in a velocity reduction of the complex. We conclude that DCX negatively regulates dynein-mediated retrograde transport through two critical interactions by regulating dynein binding to MTs and regulating the composition of the dynein motor complex.

## Editor's evaluation

In their article, Fu, Rao et al. explore the mechanisms by which the microtubule-associated protein, doublecortin (DCX), functions in regulating retrograde transport in neurons. They reconstitute a dynein-dynactin-JIP3 complex, validating JIP3 as a bona fide adaptor, and find that DCX disrupts the initial on-rate and therefore transport of this processive complex both in vivo and in vitro. This mechanism will be valuable in understanding how mutations in DCX cause lissencephaly, and this solid article will be of interest to those in the cytoskeletal and neurobiology fields.

## Introduction

Lissencephaly is a cortical malformation characterized by a 'smooth cortex' that arises from the disruption of normal development (*des Portes et al., 1998*; *Deuel et al., 2006*; *Francis et al., 1999*; *Gleeson et al., 1998*). Patients with lissencephaly often have an associated microcephaly indicating defects in neural progenitor proliferation (*Lee et al., 2010*; *Pramparo et al., 2010*). Furthermore, the

'smooth' brain is the result of abnormal neuronal migration, which causes an abnormally thick four-layered cortex (*Dobyns et al., 1984*; *Jellinger and Rett, 1976*), and many patients with lissencephaly also have a reduction or absence of major axon tracts, indicating problems with axon guidance or outgrowth (*Kappeler et al., 2007*). Thus, causative genes for lissencephaly encode proteins with critical roles in each of these steps in development: neural progenitor proliferation, neuronal migration, and axon outgrowth. Defining the molecular and cellular functions of lissencephaly genes is therefore critical for understanding early human neural development.

Interestingly, many of the causative genes for lissencephaly encode proteins related to the microtubule (MT) cytoskeleton. These include doublecortin (DCX), an MT-binding protein (*des Portes et al., 1998*; *Gleeson et al., 1998*); tubulin α1a, a major subunit of MTs (*Keays et al., 2007*); cytoplasmic dynein (hereinafter, 'dynein') heavy chain (DHC), the main retrograde motor protein (*Poirier et al., 2013*); and the dynein co-factor, lissencephaly1 (Lis1) (*Reiner et al., 1993*). This suggests that these MT proteins may be functionally related during neuronal development. Indeed, DCX overexpression can rescue nucleus–centrosome coupling defect and neuronal migration defect caused by the disruption of dynein/Lis1 function in mouse cerebellar granule neurons (*Tanaka et al., 2004*). However, the molecular mechanisms through which these MT-related proteins are functionally related are only partly understood.

Both DCX and doublecortin-like kinase 1 (DCLK1) regulate MT-based motor transport mediated by the kinesin-3 family motor KIF1A (*Deuel et al., 2006*; *Lipka et al., 2016*; *Liu et al., 2012*). As DCX binds the surface of the MT lattice (*Bechstedt and Brouhard, 2012*; *Fourniol et al., 2010*), it is logical to hypothesize that DCX regulates axonal transport by modifying the interactions of molecular motors with MTs as they step along the MT lattice. DCX was found to be a part of the dynein motor complex (*Li et al., 2021*; *Tanaka et al., 2004*) and influences the association between dynein and c-Jun NH2-terminal kinase (JNK)-interacting protein-3 (JIP3), an adaptor protein of kinesin and dynein that mediates both anterograde and retrograde transport (*Arimoto et al., 2011*; *Drerup et al., 2013*; *Li et al., 2021*), implying that DCX may influence other aspects of dynein function.

In the absence of MTs, dynein assumes an autoinhibited 'inverted' conformation (*Torisawa et al., 2014*; *Toropova et al., 2017*) and upon binding to its largest co-factor dynactin, together with a cargo adaptor such as Bicaudal D2 (BicD2), converts into a parallel conformation capable of binding MTs (*Chowdhury et al., 2015*; *McKenney et al., 2014*; *Olenick et al., 2016*; *Schlager et al., 2014*; *Zhang et al., 2017*). Dynein-dynactin-BicD2 (DDB) complex formation is facilitated by Lis1, which interacts with dynein's motor domain and prevents its autoinhibitory conformation (*Marzo et al., 2020*). However, whether dynein and dynactin can form an active motor complex with JIP3 remains unknown.

In this study, we show that DCX plays critical roles in dynein-mediated retrograde transport in axons through two different mechanisms: first, DCX decreases dynein binding to MTs, and second, DCX regulates the association of dynein with JIP3. We further demonstrate for the first time the formation of ultraprocessive dynein-dynactin-JIP3 (DDJ) motor complexes with up to two dyneins and show that DCX displaces the second dynein from a DDJ complex, resulting in a reduction of the velocity of the DDJ motor complex. Together, we demonstrate that DCX plays key roles in axon-based transport to mediate the highly specific trafficking of proteins in both anterograde and retrograde directions during neuronal growth and development by modulating the activity of MT-plus and minus-end-directed motor proteins.

## Results

### Dynein-mediated retrograde trafficking increases in the absence of DCX

Our previous work showed that DCX is essential for the function of the MT plus-end-directed kinesin-3 motor KIF1A and regulates its anterograde trafficking (*Liu et al., 2012*). DCX has also been shown to associate with the MT minus-end-directed motor dynein (*Tanaka et al., 2004*). In addition, we recently reported that dynein is involved in the DCX-mediated trafficking of Golgi extensions into dendrites, suggesting a functional link between DCX and dynein (*Li et al., 2021*). To determine whether functional interactions between DCX and dynein exist, we tested whether DCX affects the retrograde trafficking of dynein.

To visualize dynein function in vivo, we first transfected either WT, $Dcx^{-/y}$; or $Dcx^{-/y};Dclk^{-/-}$ dissociated cortical neuronal cultures with a construct expressing RFP-tagged, neuron-specific dynein intermediate chain (DIC) isoform IC-1B on days in vitro (DIV) 6 (*Ha et al., 2008*). Time-lapse imaging of DIC IC-1B-RFP was performed on DIV8 to visualize dynein motor activity directly in axons. Recorded images were converted to kymographs (*Figure 1A*). For all calculations and measurements of dynein-mediated movement, DIC above an intensity threshold located in the proximal region of axons (~100 μm away from cell body) were analyzed. A complex was counted as mobile only if the displacement was at least 5 μm over the course of the 180 s; otherwise, it was counted as stationary. Distribution calculations of DIC mobility status (anterograde, retrograde, and stationary) demonstrate that mobile DIC predominantly display retrograde movements in axons, and the percentages or run frequency of moving dynein complexes are similar in different neurons (*Figure 1B*). Remarkably, both the run length (the average distance traveled during the recorded time period) and the velocity of the fluorescently tagged dynein complexes were significantly increased in both $Dcx^{-/y}$ and $Dcx^{-/y};Dclk1^{-/-}$ axons compared with WT axons (*Figure 1C*, *Figure 1—videos 1 and 2*; run length and velocity distributions of retrograde moving DIC in different neurons are shown in *Figure 1—figure supplement 1A*). Reintroduction of DCX fully rescued the retrograde trafficking of DIC observed in $Dcx^{-/y}$ neurons (*Figure 1C*, p is 0.57 and 0.18 for DCX rescue compared with WT for DIC run length and speed, respectively). In contrast, DCLK1, a DCX domain-containing protein that is structurally similar to DCX, only partially rescued the phenotype (*Figure 1C*, p is 0.07 and 0.25 for DCLK1 rescue compared with $Dcx^{-/y}$ for DIC run length and speed, respectively).

To determine whether the dynein motility changes we see in neurons can also be observed for a physiologically relevant dynein cargo, we tested the retrograde trafficking of tropomyosin receptor kinase B (TrkB), the neurotrophin receptor whose retrograde transport is mediated by dynein (*Ha et al., 2008*; *Heerssen et al., 2004*; *Yano et al., 2001*; *Zhou et al., 2012*). As with IC-1B, the run length and velocity of retrogradely moving TrkB are also significantly increased in $Dcx^{-/y}$ axons (*Figure 1D and E*, *Figure 1—videos 3 and 4*), and reintroduction of DCX into $Dcx^{-/y}$ neurons rescued the phenotype (*Figure 1E*, p is 0.64 and 0.057 for DCX rescue compared with WT for TrkB run length and speed, respectively). The run length and velocity distributions of retrogradely transported TrkB vesicles were also calculated and are shown in *Figure 1—figure supplement 1D*. Like IC-1B, the majority of mobile TrkB vesicles in axons were transported in retrograde direction and no significant differences were found between the percentage of vesicles measured under WT, $Dcx^{-/y}$, and rescue conditions for anterograde, retrograde moving particles, and stationary vesicles (*Figure 1—figure supplement 1B*). About 10% of the imaged TrkB vesicles were transported in anterograde direction, and we analyzed those TrkB vesicles to determine whether DCX has effects on anterograde TrkB transport. We observed no significant differences in both velocity and run length of anterogradely transported TrkB vesicles between WT and $Dcx^{-/y}$ neurons (*Figure 1—figure supplement 1C*). Overall, our data indicate that loss of DCX increases dynein-mediated vesicular retrograde transport in the axon.

## The effect of DCX on retrograde transport is mediated through interactions between DCX and dynein

In addition to its binding to MTs, DCX also associates with the dynein motor complex (*Tanaka et al., 2004*; *Taylor et al., 2000*). To further probe the interaction of DCX with dynein, we used recombinant HA (negative control) and HA-DCX constructs as bait to pull down DCX-interacting proteins from mouse brains using mass-spectrometry analysis. *Table 1* shows that our analysis identified cytoplasmic dynein 1 intermediate chain 2 and dynactin subunit 5 as DCX-interacting proteins. Although this analysis does not rule out the possibility that other unknown proteins are required for the interaction of DCX with IC2 and the dynactin subunit 5, it is very likely that dynein and DCX interact directly as DCX affects the composition of DDJ motor complexes that are formed from recombinantly expressed proteins (see below).

To define which domain of DCX is critical for its association with the dynein motor complex, we expressed HA-tagged full-length DCX (FL-DCX), an N-terminal DCX construct (N-DCX) containing the R1 and R2 MT-binding domains (amino acids 1–270), or a C-terminal construct containing the serine/proline (SP)-rich domain of DCX (C-DCX) (amino acids 271–361, *Figure 2A*) in HEK293 cells. Consistent with the previous report (*Tanaka et al., 2004*), DIC precipitated with FL-DCX (*Figure 2B*). Interestingly, more DIC precipitated with N-DCX than FL-DCX (*Figure 2B*); similarly, N-DCX showed

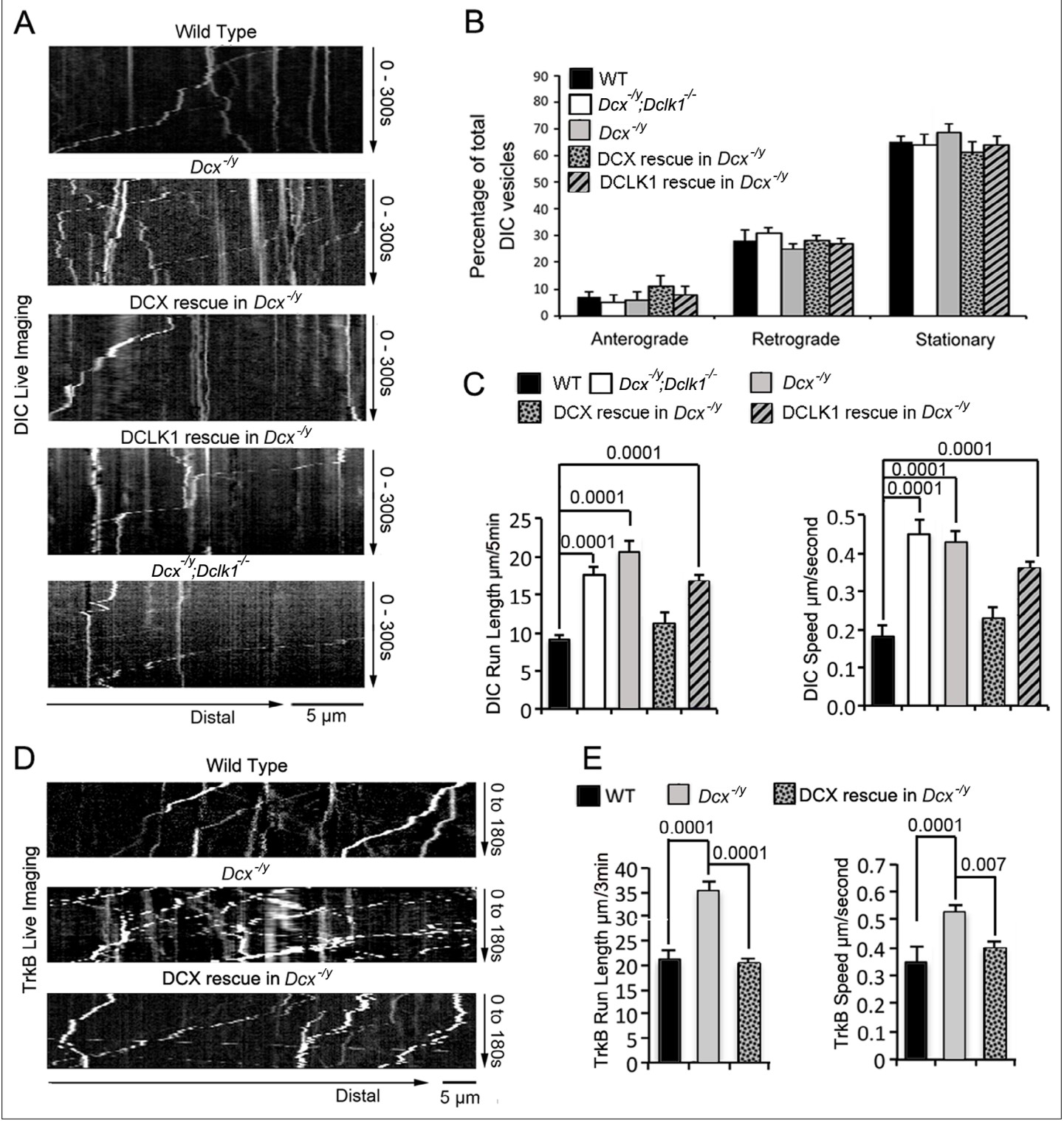

**Figure 1.** The retrograde trafficking of the dynein motor and TrkB transport is increased in axons without doublecortin (DCX). (**A**) WT, *Dcx⁻/ʸ* or *Dcx⁻/ʸ;Dclk1⁻/⁻* associated cortical neuronal culture were transfected with plasmids expressing DIC-RFP on days in vitro (DIV) 6 and imaged on DIV8. For rescue experiments, *Dcx⁻/ʸ* neurons were transfected with plasmids expressing DIC-RFP combined with plasmids expressing either DCX-GFP or DCLK1-GFP. Representative kymographs of DIC-RFP transport in axons are shown. (**B**) Distribution calculations of the DIC vesicle mobility status (anterograde, retrograde, and stationary) are demonstrated. No significant differences are observed among different neurons. (**C**) Quantifications of DIC-RFP run length (mean+/- SEM) within 300 s and velocity (mean+/-SEM) are shown. DCX, but not DCLK1, fully rescued the increased dynein motor transport observed in DCX-deficient axons. p-Values comparing WT and DCX rescue for run length and speed are 0.57 and 0.18, respectively. Other p-values are

*Figure 1 continued on next page*

*Figure 1 continued*

shown in the figure. (**D**) Dissociated cortical neuronal cultures from WT or *Dcx^-/y* mice were transfected with plasmids expressing TrkB-RFP with/without plasmids expressing DCX-GFP on DIV6 and imaged on DIV8. Representative kymographs of TrkB-RFP transport in axons are shown. (**E**) Quantification of vesicle run length (mean+/- SEM) within 180 s and velocity (mean+/- SEM) are demonstrated. DCX rescued the increased TrkB-RFP transport in DCX-deficient axons. Data are based on three independent experiments of each condition. Total numbers of neurons (N) and vesicles (V) used in the calculations are indicated in *Figure 1—figure supplement 1*.

The online version of this article includes the following video and figure supplement(s) for figure 1:

**Figure supplement 1.** Run length and velocity distributions of dynein intermediate chain (DIC) and TrkB in different neurons.

**Figure 1—video 1.** Live-cell imaging shows dynein intermediate chain (DIC) mobility in a WT neuron.

https://elifesciences.org/articles/82218/figures#fig1video1

**Figure 1—video 2.** Live-cell imaging shows dynein intermediate chain (DIC) mobility in a *Dcx^-/y* neuron.

https://elifesciences.org/articles/82218/figures#fig1video2

**Figure 1—video 3.** Live-cell imaging shows TrkB mobility in a WT neuron.

https://elifesciences.org/articles/82218/figures#fig1video3

**Figure 1—video 4.** Live-cell imaging shows TrkB mobility in a *Dcx^-/y* neuron.

https://elifesciences.org/articles/82218/figures#fig1video4

an increased immunoprecipitation of the DHC compared with FL-DCX (*Figure 2—figure supplement 1A*). This result suggests that N-DCX has a stronger affinity for the dynein motor complex than FL-DCX. We reasoned that if the interaction of DCX with dynein plays an important role in regulating dynein function, then N-DCX should have a stronger effect on regulating dynein-mediated retrograde transport than FL-DCX. Indeed, introducing N-DCX either into DCX knockout neurons (*Figure 2C and D*, *Figure 2—figure supplement 2A and B*) or WT neurons (*Figure 2F and G*, *Figure 2—figure supplement 2C and D*) decreases the retrograde transport of TrkB to a greater extent than FL-DCX. Our results suggest that DCX decreases dynein-mediated retrograde transport through interactions with dynein motor complex.

## The C-terminal S/P-rich domain of DCX decreases DCX-dynein interactions

In contrast to FL-DCX and N-DCX, C-DCX did not immunoprecipitate with dynein (*Figure 2—figure supplement 1B and C*), consistent with previous report (*Taylor et al., 2000*). Since N-DCX, which misses the C-terminal domain, has a stronger affinity for dynein than FL-DCX, we hypothesized that the C-terminus of DCX inhibits the interaction between DCX and dynein. Indeed, in the presence of C-DCX, significantly less DIC precipitated with FL-DCX (*Figure 2E*); similarly, less DHC was precipitated with either FL-DCX or N-DCX in the presence of C-DCX (*Figure 2—figure supplement 2B and C*). Furthermore, C-DCX overexpression in WT neurons significantly increased dynein-mediated retrograde transport of TrkB (*Figure 2F and G*, *Figure 2—figure supplement 2C and D*). Taken together, these data indicate that DCX decreases dynein-mediated retrograde transport through interactions with the dynein motor complex through its N-terminus, while the C-terminal domain of DCX negatively impacts this interaction to influence dynein-based cargo trafficking.

## The effects of DCX on retrograde trafficking require DCX-MT interactions

Previous work has shown that the binding of DCX to MTs contributes to DCX's cellular functions (*Moslehi et al., 2019*; *Reiner, 2013*; *Schaar et al., 2004*; *Yap et al., 2012*) and that DCX's MT interactions occur cooperatively (*Bechstedt and Brouhard, 2012*). To test whether MT binding and the underlying cooperativity of DCX's MT interactions play a role in regulating dynein-based vesicular transport, we tested whether two DCX mutants, DCX A71S and T203R, could rescue the increase in retrograde transport of TrkB vesicles in DCX knockout neurons. These mutations, located in the R1 and R2 region of DCX, respectively, cause lissencephaly in humans and have been shown to decrease the cooperative MT binding of DCX (*Bechstedt and Brouhard, 2012*). Importantly, these mutations have no effect on DCX's ability to associate with dynein in vitro (*Figure 2—figure supplement 1D*). Both mutants were unable to rescue the phenotype of increased retrograde transport of TrkB (*Figure 3A*, *Figure 3—figure supplement 1*), suggesting that the cooperative binding of DCX to MTs is required

**Table 1.** Pull-down assay results show cytoskeleton proteins associated with doublecortin (DCX). Additional results from pull-down assay can be found in the Dryad Digital Repository at https://doi.org/10.5061/dryad.vmcvdncwt.

| Protein IDs | Protein names | Q-value | Score | Average normalized intensity | |
| --- | --- | --- | --- | --- | --- |
| | | | | Control (**HA**) | HA-DCX |
| Q61301 | Catenin alpha-2 | 0 | 83 | 0 | 158960000 |
| P57780 | Alpha-actinin-4 | 0 | 76 | 0 | 246655000 |
| Q8BMK4 | Cytoskeleton-associated protein 4 | 0 | 70 | 0 | 124115000 |
| Q02248 | Catenin beta-1 | 0 | 55 | 0 | 133995000 |
| P05213 | Tubulin alpha-1B chain; tubulin alpha-4A chain | 0 | 52 | 0 | 600270000 |
| Q9D6F9 | Tubulin beta-4A chain | 0 | 38 | 0 | 91708000 |
| Q8BTM8 | Filamin-A | 0 | 38 | 0 | 48710000 |
| Q8K341 | Alpha-tubulin N-acetyltransferase 1 | 0 | 36 | 0 | 9883000 |
| Q99KJ8 | Dynactin subunit 2 | 0 | 32 | 0 | 67364000 |
| Q3TPJ8 | Cytoplasmic dynein 1 intermediate chain 2 | 0 | 25 | 0 | 34566000 |
| Q9CPW4 | Actin-related protein 2/3 complex subunit 5 | 0 | 21 | 0 | 32742000 |
| Q9D898 | Actin-related protein 2/3 complex subunit 5-like protein | 0 | 17 | 0 | 3859300 |
| Q6R891 | Neurabin-2 | 0 | 15 | 0 | 6523000 |
| P28667 | MARCKS-related protein | 0 | 14 | 0 | 52599500 |
| Q9JM76 | Actin-related protein 2/3 complex subunit 3 | 0 | 14 | 0 | 30507500 |
| Q7TPR4 | Alpha-actinin-1 | 0 | 12 | 0 | 6880500 |
| P60710 | Actin, cytoplasmic 1, N-terminally processed | 0 | 9 | 0 | 30897500 |
| Q3UX10 | Tubulin alpha chain-like 3 | 0 | 8 | 0 | 7851000 |
| Q9CQV6 | Microtubule-associated proteins 1A/1B light chain 3B | 0 | 7.5 | 0 | 42111500 |
| Q922F4 | Tubulin beta-6 chain | 0.001 | 7.3 | 0 | 12908500 |
| Q9QZB9 | Dynactin subunit 5 | 0.007 | 6.2 | 0 | 2061750 |

The online version of this article includes the following source data for table 1:

**Source data 1.** Full list of proteins identified from pull-down assay.

for the DCX-induced decrease in dynein-based retrograde transport. In addition, our MT-binding assay demonstrates that, while C-DCX itself does not bind MTs, C-DCX increases the interactions of DCX with MTs (*Figure 3B*). This suggests that the interactions between DCX and MTs are enhanced by DCX's C-terminal domain, which is consistent with recent findings that the tail of DCX (amino acid 303 to the C-terminal end) helps to maintain the associations between DCX molecules on the MT lattice (*Rafiei et al., 2022*).

## DCX decreases dynein association with MTs

Since DCX enhances the binding of KIF1A to MTs and regulates KIF1A-mediated transport (*Liu et al., 2012*), we tested whether DCX also alters dynein's interactions with MTs by performing an MT-binding assay using brain lysate from either WT, *Dcx*$^{-/y}$, or *Dcx*$^{-/y}$;*Dclk1*$^{-/-}$ mice. Our results show that significantly more DIC protein precipitates with MTs in the absence of DCX (*Figure 3C and D*). Therefore,

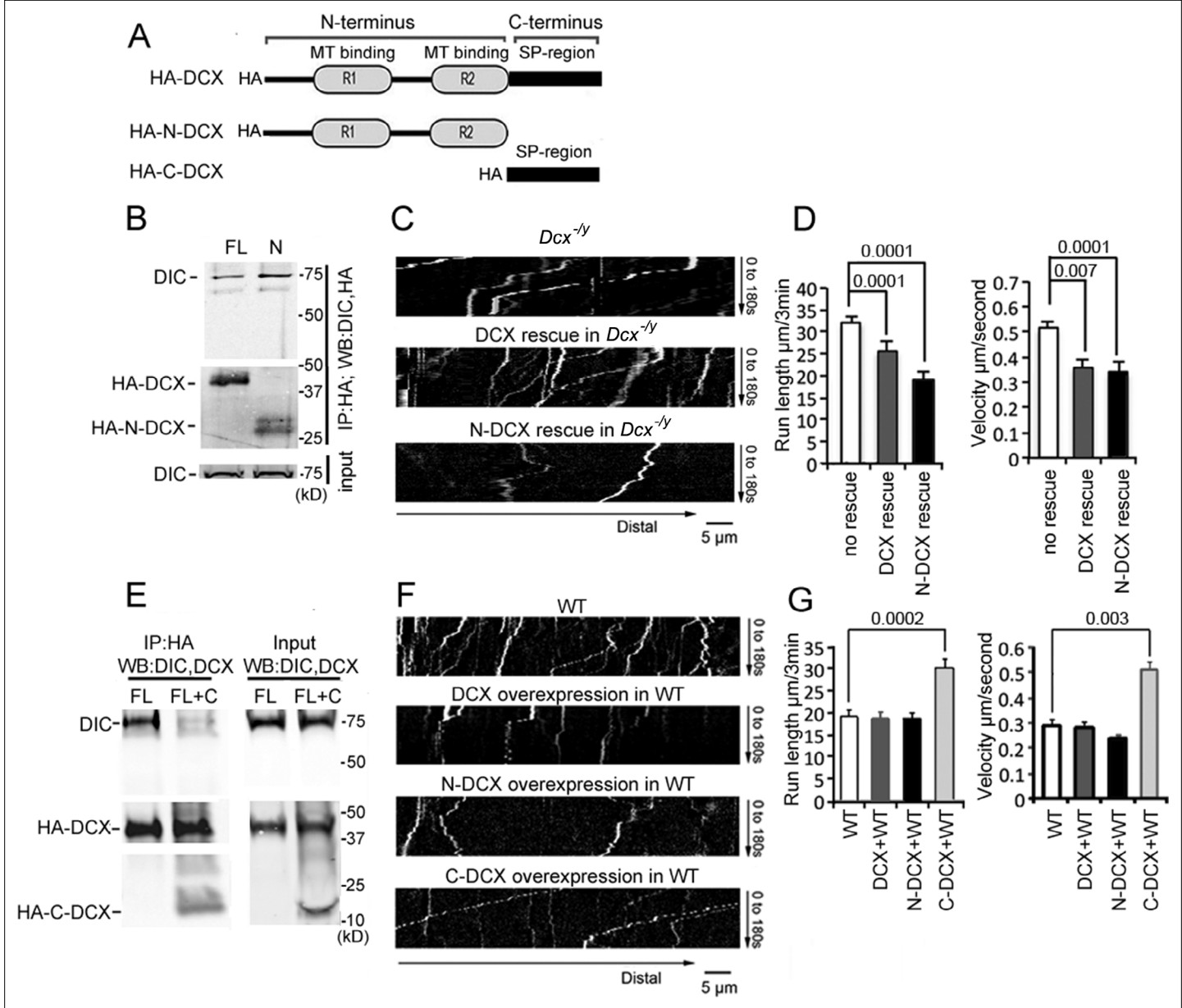

**Figure 2.** Doublecortin (DCX) affects the retrograde transport through DCX/dynein interaction. (**A**) A schematic of DCX protein domain structure. N-DCX has N-terminal R1 and R2 domains representing microtubule (MT)-binding domains of DCX, while C-DCX has DCX C-terminal serine/proline (SP) rich domain. (**B**) More N-DCX proteins are pulled down with DIC compared with full-length DCX. HEK293 cells were transfected with plasmids expressing HA-tagged either full-length DCX (FL) or N-DCX (N) for 2 days. Protein lysates were used for immunoprecipitation using antibodies for HA and analyzed by Western blot for DIC and HA. (**C**) Dissociated cortical neuronal culture from *Dcx*<sup>-/y</sup>;*Dclk1*<sup>-/-</sup> mice were transfected with plasmids expressing TrkB-RFP with/without plasmids expressing DCX or N-DCX on days in vitro (DIV)6 and imaged on DIV8. Representative kymographs of TrkB-RFP transports in axons are shown. (**D**) Quantifications of TrkB-RFP run length (mean+/- SEM) within 180s and velocity (mean+/-SEM) are shown. The expression of either full-length DCX or N-DCX in DCX knockout neurons significantly decreased TrkB retrograde transport while N-DCX has stronger effect compared with full-length DCX. (**E**) C-DCX decreases DCX/DIC association. HEK293 cells were transfected with plasmids expressing HA-tagged full-length DCX (FL) with/without plasmid expressing C-DCX for 2 days. Protein lysates were used for immunoprecipitation using antibodies for HA and analyzed by Western blot for DIC and HA. (**F**) Dissociated cortical neuronal culture from wild-type mice were transfected with plasmids expressing TrkB-RFP with/without plasmids expressing DCX, N-DCX, or C-DCX on DIV6 and imaged on DIV8. Representative kymographs of TrkB-RFP transports in axons are shown. (**G**) Run length within 180 s and velocity distributions of retrograde TrkB complexes in axons are quantified. C-DCX overexpression in wild-type neurons mimicked the phenotype of TrkB retrograde trafficking observed in *Dcx*<sup>-/y</sup> axons. All quantification data are based on three independent experiments of each condition. p-Values from *t*-tests are shown in each panel. Total numbers of neurons (N) and vesicles (V) used in the calculations are indicated in *Figure 2—figure supplement 2*. See also *Figure 2—figure supplement 1*.

*Figure 2 continued on next page*

*Figure 2 continued*

The online version of this article includes the following source data and figure supplement(s) for figure 2:

**Source data 1.** Uncropped WB gels for *Figure 2*.

**Figure supplement 1.** Doublecortin (DCX) associates with dynein heavy chain through its N-terminal domain.

**Figure supplement 1—source data 1.** Uncropped WB gels for *Figure 2—figure supplement 1*.

**Figure supplement 2.** Doublecortin (DCX) affects the retrograde transport through DCX/dynein interaction.

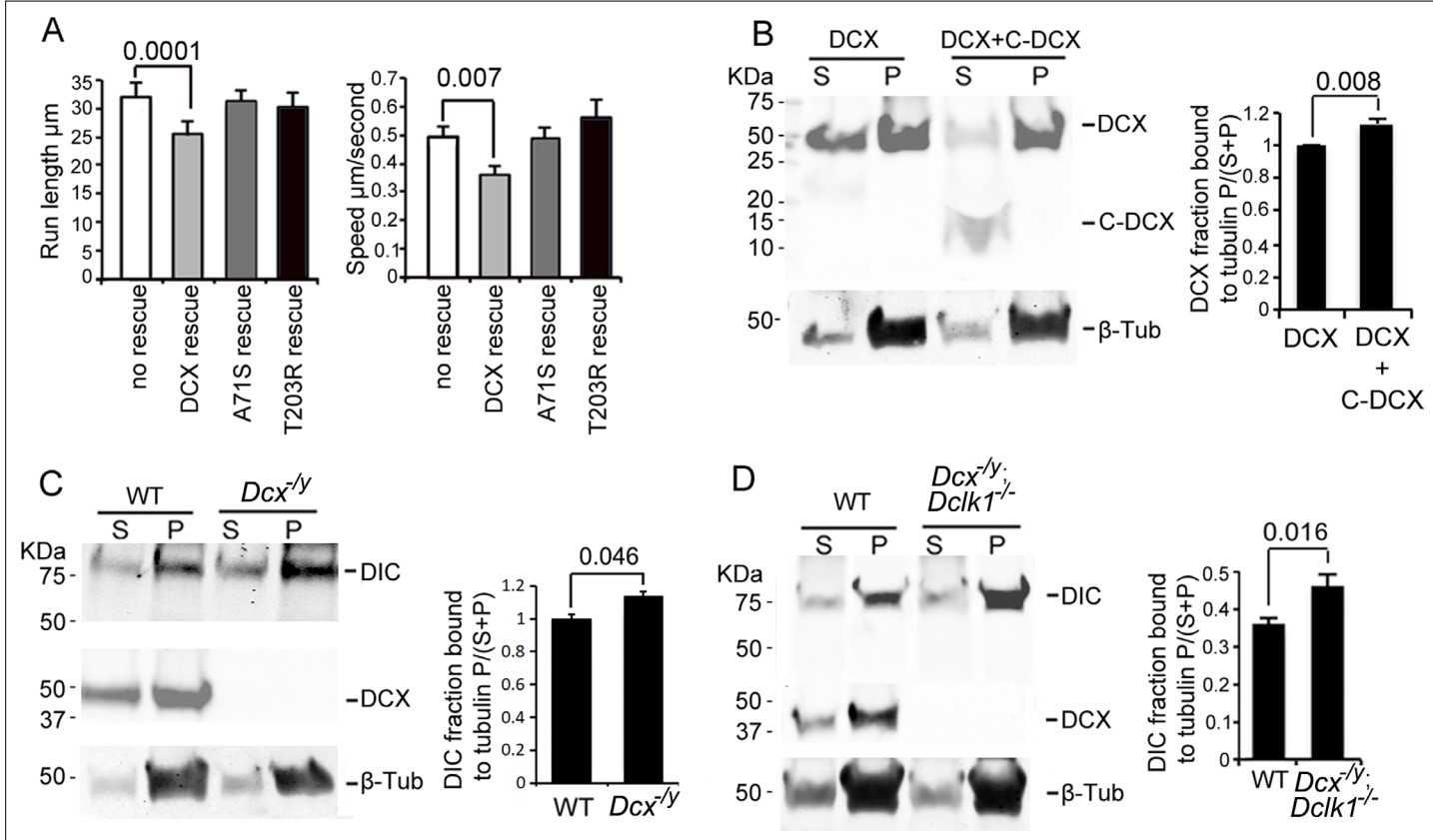

**Figure 3.** Doublecortin (DCX) association with microtubules (MTs). (**A**). Dissociated cortical neuronal cultures from WT or *Dcx⁻/y* mice were transfected with plasmids expressing TrkB-RFP with/without plasmids expressing DCX-GFP, DCXA71S, or DCXT203R on days in vitro (DIV)6 and imaged on DIV8. Quantification of Run length (mean +/- SEM) within 180 s and velocity (mean+/-SEM) is demonstrated. DCX, but not DCXA71S or DCXT203R, rescued the increased TrkB-RFP transport in DCX-deficient axons. All quantifications are based on three independent experiments of each condition. p-Values from *t*-tests are shown in each panel. Total numbers of neurons (N) and vesicles (V) used in the calculations are indicated in *Figure 3—figure supplement 1*. (**B**). Protein lysate from HEK293 cells expressing HA-DCX or HA-DCX plus HA-C-DCX are incubated with exogenously added MTs, which are then pelleted by ultracentrifugation. Western blot for HA in supernatant (S) or pellet (P) is performed to determine the amount of DCX or C-DCX associated with MTs. Representative Western blots are shown. Fraction of DCX bound to tubulin is calculated (tubulin bound = P/(S + P)) and compared. Significantly more DCX is bound to MTs in the presence of C-DCX. p-Value from *t*-test is shown. (**C, D**) Brain lysate from P0 WT, *Dcx⁻/y* or *Dcx⁻/y;Dclk1⁻/⁻* mice are incubated with exogenously added MTs, which are then pelleted by ultracentrifugation. Polymerized MTs are in the pellet. Western blot of dynein intermediate chain (DIC) in supernatant (S) or pellet (P) is performed to determine the amount of DIC associated with MTs. Representative Western blots are shown. Fraction of DIC bound to tubulin is calculated (tubulin bound = P/(S + P)) and compared between WT and *Dcx⁻/y*; or WT and *Dcx⁻/y;Dclk1⁻/⁻*. Significantly more DIC is bound to MTs in the absence of DCX. p-Value from *t*-test is shown. All quantifications are based on three independent experiments.

The online version of this article includes the following source data and figure supplement(s) for figure 3:

**Source data 1.** Uncropped WB gels for *Figure 3*.

**Figure supplement 1.** Doublecortin (DCX) effect on the retrograde trafficking needs DCX/microtubule (MT) interaction.

in contrast to DCX's positive effects on KIF1A's association with MTs (*Liu et al., 2012*), DCX decreases dynein-MT interactions.

## DCX negatively regulates dynein-mediated retrograde transport by regulating JIP3 association with dynein

Various binding partners are known to regulate dynein-based cargo transport (*Vallee et al., 2012*), and DCX could achieve its effect on the retrograde transport of TrkB by altering dynein's association with regulatory proteins. Indeed, we previously reported that DCX regulates the interaction of dynein with its cargo adaptor JIP3 (*Li et al., 2021*), suggesting that DCX may regulate dynein-based cargo transport by controlling dynein-dynactin complex assembly and/or dynein-cargo attachment. In support of this idea, JIP3 also decreases the binding of DCX to dynein (*Figure 4A*). As JIP3 serves as an adaptor protein for both kinesin and dynein (*Arimoto et al., 2011*; *Drerup et al., 2013*), DCX could differentially regulate anterograde vs. retrograde transport by competing with JIP3 for the binding of dynein. To test this possibility, we examined the retrograde transport of TrkB-RFP in *Dcx* knockout cortical neurons transfected with either control shRNA or JIP3 shRNA. Knockdown of JIP3 indeed significantly decreases the retrograde transport of TrkB (*Figure 4B and C*, *Figure 4—figure supplement 1*, *Figure 4—video 1*) while overexpression of JIP3 in WT neurons increases the retrograde transport TrkB (*Figure 4B and C*, *Figure 4—figure supplement 1*, *Figure 4—video 2*). Based on these results, we conclude that at least two mechanisms are at play when DCX regulates dynein-based transport: first, through negatively regulating dynein interactions with MTs, and secondly, through negatively regulating dynein interactions with JIP3.

## Dynein, dynactin, and JIP3 form a processive tripartite complex in vitro

To directly determine how DCX affects the motion of the dynein motor complex, we used total internal reflection fluorescence (TIRF) microscopy and performed single-molecule motility studies using purified components. Dynein, which assumes an autoinhibited conformation in isolation (*Torisawa et al., 2014*; *Zhang et al., 2017*), moves processively along coverslip-attached MTs after its activation through the formation of a complex with its largest cofactor dynactin and a coiled-coil cargo adaptor protein such as Bicaudal D2 (BicD2) (*McKenney et al., 2014*; *Splinter et al., 2012*). Complexes such as dynein-dynactin-BicD2 (DDB), dynein-dynactin-BicDR1 (Bicaudal-D-related protein 1) (DDR), and dynein-dynactin-Hook3 (DDH), which have been recently shown to bind up to two dyneins (*Grotjahn et al., 2018*; *Urnavicius et al., 2018*), have been extensively studied using single-molecule TIRF assays (*Christensen et al., 2021*; *McClintock et al., 2018*; *McKenney et al., 2014*; *Sladewski et al., 2018*; *Urnavicius et al., 2018*). However, whether dynein-dynactin-JIP3 (DDJ) motor complexes can be reconstituted in vitro remains unknown.

Previous biochemical studies have shown that JIP3 interacts with dynein's light intermediate chain (LIC) (*Arimoto et al., 2011*) and with kinesin-1's light (*Bowman et al., 2000*) and heavy chains (*Sun et al., 2011*). Mutations in JIP3 result in the mis-localization of the dynein and impair retrograde transport (*Arimoto et al., 2011*; *Celestino et al., 2022*). However, the current consensus is that the coiled-coil region of JIP3 is too short to form a stable complex with dynein and dynactin (*Chaaban and Carter, 2022*; *Lee et al., 2020*; *Reck-Peterson et al., 2018*). Indeed, the putative N-terminal α-helix of JIP3, which is followed by an intrinsically disordered domain, extends only up to ~180 residues according to the structural prediction by AlphaFold (*Jumper et al., 2021*; *Figure 5—figure supplement 1C*). In contrast, cargo adaptors that have been successfully used for in vitro motility assays have been predicted to have substantially longer N-terminal coiled-coil domains. For example, the coiled coil of BicD2 is predicted to extend to ~270 residues and cover the full shoulder of dynactin (*Figure 5—figure supplement 1A and B*), in agreement with cryo-EM studies (*Urnavicius et al., 2015*). Nonetheless, based on our in vivo and immunoprecipitation results, we hypothesized that JIP3 can form a tripartite complex with dynein and dynactin, possibly through a transition of its disordered domain into a more ordered conformation upon binding to dynein/dynactin. A similar disorder-to-order transition has been recently reported for Nup358, which changed a random coil into an α-helix upon binding to BicD2 (*Gibson et al., 2022*).

To test whether JIP3 can form an active DDJ motor complex, we generated and expressed a mouse JIP3 construct containing the N-terminal coiled-coil and the predicted adjacent intrinsically disordered domain (amino acids 1–240) in *Escherichia coli* (*Figure 5A*, *Figure 5—figure supplement*

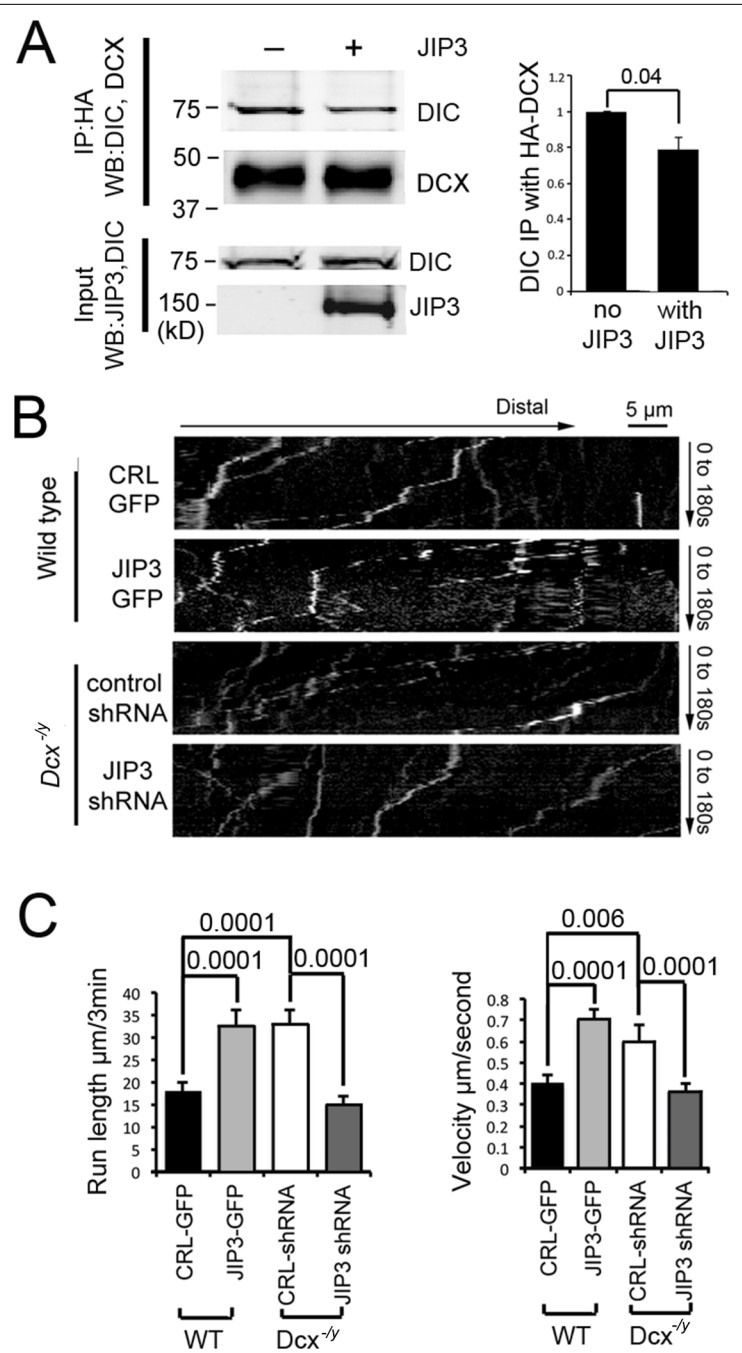

**Figure 4.** Doublecortin (DCX) and JIP3 competitively bind to the dynein motor complex, and JIP3 enhances retrograde transport mediated by dynein. (**A**) The presence of JIP3 decreases the interaction between DCX and dynein intermediate chain (DIC). HEK293 cells were transfected with plasmids expressing neuron-specific dynein intermediate chain isoform IC-1B and HA-tagged DCX with or without JIP3. Antibody for HA was used to precipitate HA-DCX and associated proteins. Western blot analysis of DIC and HA was performed to detect DIC immunoprecipitated with DCX. In the presence of JIP3, less DIC was associated with DCX, while total protein amount of either HA-DCX or DIC in the lysate was the same. Quantification of DIC bands of Western blot results (three independent experiments) after IP with HA were calculated and normalized with DIC levels in the lysate. p-Value from *t*-test is shown. (**B**) Cultured cortical neurons from P0 WT mouse brains were transfected with plasmids expressing TrkB-RFP with or without JIP3-GFP. Neurons from *Dcx⁻ʸ* mouse brains were transfected with plasmids expressing TrkB-RFP with or without JIP3 shRNA. Representative kymographs of TrkB-RFP trafficking are demonstrated. (**C**) Quantification of TrkB run length and velocity. Overexpression of JIP3 significantly increases

*Figure 4 continued on next page*

*Figure 4 continued*

run length and velocity of TrkB in WT neurons. Downregulation of JIP3 by shRNA in *Dcx⁻/ʸ* neurons decreases TrkB retrograde transport. p-Values from *t*-tests are shown. All quantification data are based on three independent experiments of each condition. p-Values from *t*-tests are shown in each panel. Total numbers of neurons (N) and vesicles (V) used in the calculations are indicated in *Figure 4—figure supplement 1*. See also *Figure 4—video 1* and *Figure 4—video 2*.

The online version of this article includes the following video, source data, and figure supplement(s) for figure 4:

**Source data 1.** Uncropped WB gels for *Figure 4*.

**Figure supplement 1.** JIP3 enhances retrograde transport of TrkB.

**Figure 4—video 1.** Live-cell imaging shows TrkB mobility in a *Dcx⁻/ʸ* neuron transfected with JIP3-shRNA.
https://elifesciences.org/articles/82218/figures#fig4video1

**Figure 4—video 2.** Live-cell imaging shows TrkB mobility in a WT neuron transfected with JIP3.
https://elifesciences.org/articles/82218/figures#fig4video2

*2A*). The homolog of this construct in *Caenorhabditis elegans* has been shown to interact with dynein through the dynein LIC (*Arimoto et al., 2011*). To generate and purify full-length human dynein with its associated five subunits (IC, LIC, Tctex, LC8, and Robl), we co-expressed the five subunits with the HC of dynein in insect cells as done before (*Schlager et al., 2014*; *Figure 5—figure supplement 2C*). To allow single-molecule fluorescence imaging of both dynein and JIP3, we labeled the dynein HC with SNAP-TMR via an N-terminal SNAP-tag and JIP3 with Halo-JF646 via a C-terminal HaloTag (*Figure 5A*).

As expected for an autoinhibited motor, the purified dynein only diffused along MTs or bound rigidly (*McKenney et al., 2014*; *Schlager et al., 2014*), and addition of dynactin (purified from cow brain; *Schlager et al., 2014*) did not activate its motion (*Figure 5B*). While JIP3 transiently interacts with dynein (*Figure 5—figure supplement 3A*, *Figure 5—video 1*), it can neither stably bind to dynein (*Figure 5—figure supplement 3B*) nor activate dynein motion (*Figure 5B*), similar to other studied dynein adaptors (*McKenney et al., 2014*; *Schlager et al., 2014*). However, strikingly, when we incubated JIP3 with dynein and dynactin on ice for 1 hr in a 1:1:1 stoichiometry, DDJ complexes formed and moved processively along MTs (*Figure 5B*) at a velocity of 0.8 [0.7, 0.9] µm/s (median with 95% CIs) (*Figure 5C*), which is comparable to other dynein complexes (*Elshenawy et al., 2019*; *McKenney et al., 2014*; *Urnavicius et al., 2018*). Our results demonstrate that JIP3 and dynactin are involved in dynein activation and that JIP3 can form highly processive DDJ complexes despite its predicted short coiled-coil domain.

## DCX decreases the velocity of DDJ motor complexes

To determine whether DCX negatively impacts the velocity of DDJ motor complexes as suggested by our in vivo results, we expressed full-length DCX and N-DCX with a C-terminal ybbR-tag (*Yin et al., 2006*) for labeling with CoA-CF488 or CoA-JF549 in *E. coli* (*Figure 5—figure supplement 2B*). We chose the small α-helical ybbR tag over the commonly used GFP tag (*Bechstedt and Brouhard, 2012*; *Ettinger et al., 2016*) to reduce possible steric blocking of dynein MT-binding sites by the introduced tag. At 10 nM concentration, DCX fully decorated MTs in the dynein motility buffer (*Figure 5—figure supplement 4A*), while N-DCX had a much weaker affinity for MTs (*Figure 5—figure supplement 4B*), which is consistent with our immunoprecipitation data that demonstrated that the addition of C-DCX increased the affinity of DCX for MTs (*Figure 3B*). Only when the ionic strength of the buffer was reduced, increasing amounts of N-DCX bound to MTs (*Figure 5—figure supplement 4C*). These observations confirm that our ybbR-tagged and labeled DCX constructs are functional.

Decorating MTs using 10 nM DCX slightly reduced DDJ's velocity (0.62 [0.50, 0.70] µm/s, *p<0.1) (*Figure 5C*), while N-DCX had a stronger effect on the velocity (0.54 [0.45, 0.67] µm/s, ***p<0.001) (*Figure 5C*), which is consistent with our in vivo results (*Figure 2*). The more pronounced effect of N-DCX on DDJ velocity supports our in vivo results that showed the C-terminus of DCX negatively regulates DCX-dynein binding (*Figure 2E*). Moreover, since N-DCX does not bind MTs in our regular motility buffer unless we use motility buffer with half ionic strength (*Figure 5—figure supplement 4B and C*), N-DCX likely acts directly upon the DDJ complex rather than through MT binding.

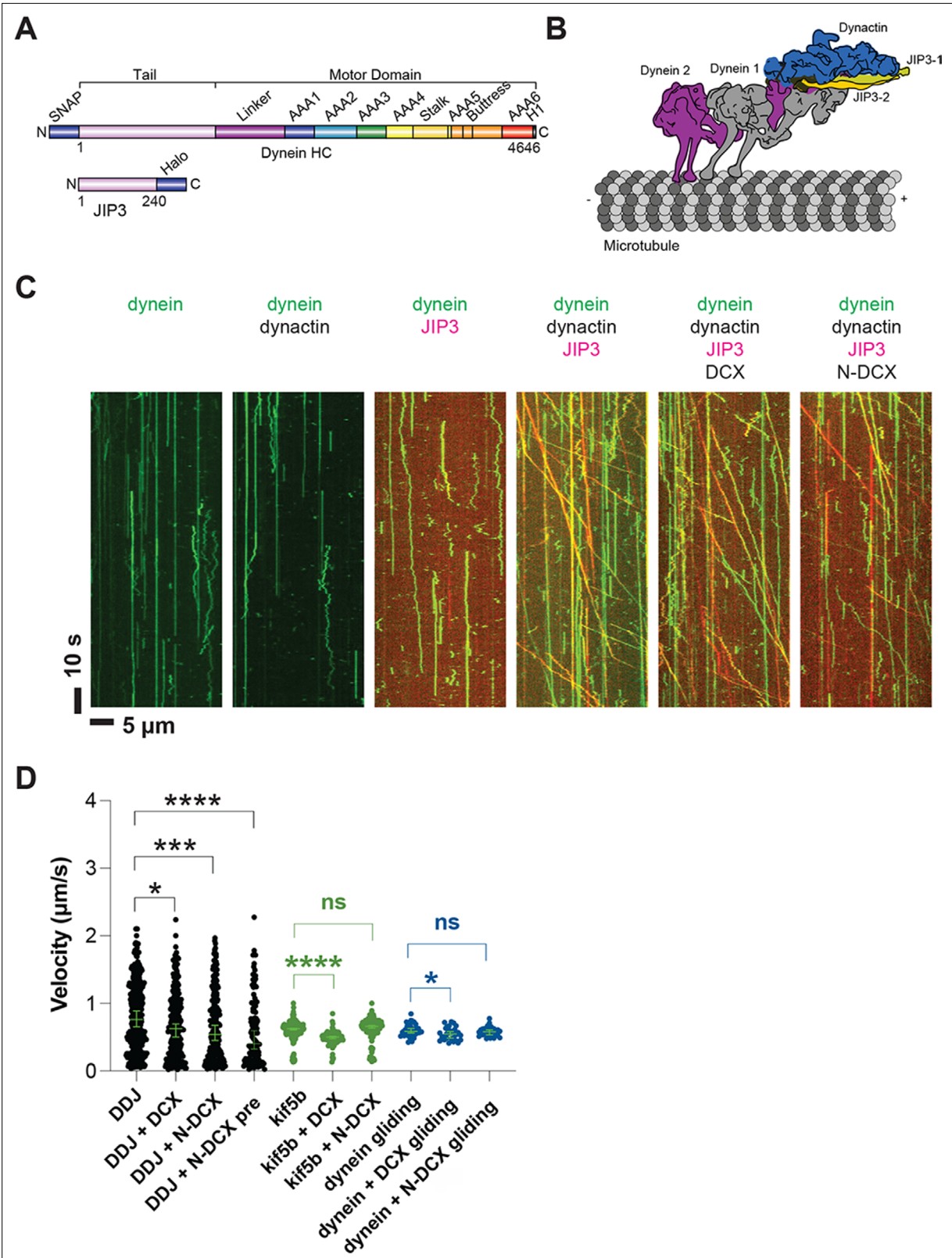

**Figure 5.** Dynein and dynactin form an active motor complex with JIP3 in vitro, and doublecortin (DCX) reduces its velocity. (**A**) Illustrations of the JIP3 and DCX constructs (left) and the DDJ motor complex (right). (**B**) Kymographs of dynein in the absence and presence of dynactin, JIP3, DCX, and N-DCX. Dynein was labeled with SNAP-TMR (green) and JIP3 was labeled with Halo-JP646 (red). (**C**) The velocity of DDJ motor complexes, KIF5B, and gliding MTs powered by surface-absorbed single-headed dynein. The green bars represent the median with 95% CI. DDJ (DDJ only): 0.76 [0.65, 0.89]

*Figure 5 continued on next page*

*Figure 5 continued*

µm/s; DDJ + DCX (DDJ with 10 nM DCX): 0.62 [0.50, 0.70] µm/s (KS test, *p<0.1); DDJ + N-DCX (DDJ with 10 nM N-DCX): 0.54 [0.45, 0.67] µm/s (KS test, ***p<0.001); DDJ + N-DCX pre (dynein, dynactin, JIP3, and N-DCX assembled in the ratio of 1:1:1:1): 0.41 [0.32, 0.60] µm/s (KS test, ****p<0.0001). kif5b (kif5b only): 0.63 [0.61, 0.63] µm/s; kif5b + DCX (kif5b with 10 nM DCX): 0.50 [0.48, 0.51] µm/s (unpaired *t*-test, ****p<0.0001); kif5b + N-DCX (kif5b with 10 nM N-DCX): 0.66 [0.64, 0.67] µm/s (unpaired *t*-test, n.s.). MT gliding (powered by single-headed human dynein): 0.59 [0.56, 0.63] µm/s; dynein + DCX (MT gliding with 10 nM DCX): 0.55 [0.48, 0.58] µm/s (unpaired *t*-test, *p<0.1); dynein + N-DCX (MT gliding with 10 nM N-DCX): 0.58 [0.53, 0.61] µm/s (unpaired *t*-test, n.s.). From left to right, n = 342, 275, 252, 115, 234, 103, 117, 33, 31, and 28. See also *Figure 5—figure supplements 1–5* and *Figure 5—video 1*.

The online version of this article includes the following video and figure supplement(s) for figure 5:

**Figure supplement 1.** Comparison of the predicted structures of BicD2 and JIP3 with cryo-EM structure of DDB containing BicD2.

**Figure supplement 2.** PAGE gels of recombinant expressed proteins.

**Figure supplement 3.** JIP3 has a transient affinity for dynein.

**Figure supplement 4.** Doublecortin (DCX) binds microtubules (MTs).

**Figure supplement 5.** Doublecortin's (DCX's) effects on the microtubule (MT) on-rate and the MT-bound time of single-headed human dynein.

**Figure 5—video 1.** JIP3 has a transient affinity for dynein.

https://elifesciences.org/articles/82218/figures#fig5video1

To determine whether N-DCX acts specifically on dynein, we also tested FL-DCX and N-DCX on another canonical MT-based motor, the kinesin-1 family member KIF5B. In contrast to the effects on DDJ, N-DCX has no effects on the velocity of KIF5B, while FL-DCX reduces the velocity of KIF5B (*Figure 5C*), a result that agrees with a recent study that showed that DCX decreases the binding of kinesin-1 to MTs (*Monroy et al., 2020*). These results collectively suggest that DCX differentially affects the velocities of DDJ and kinesin-1: while DCX affects kinesin-1 motility through its binding to MTs, N-DCX affects DDJ motility through interactions with the dynein motor complex.

To dissect which component of the dynein motor complex N-DCX regulates, we first performed an MT-gliding assay using an N-terminal GFP-tagged single-headed human dynein construct expressed in insect cells (*Htet et al., 2020*). The single-headed dynein only contains the motor domain of the heavy chain, while the tail domain and all other subunits are absent. As this motor construct is nonprocessive (*Trokter et al., 2012*), we performed an MT-gliding assay to probe the effects of DCX on the activity of the dynein motor domain. To do so, we bound the GFP-tagged dyneins to a cover-glass surface via anti-GFP antibody at a motor-surface density that supports smooth gliding of MTs along the cover-glass surface. If N-DCX acts directly on the dynein motor domain as the dynein co-factor Lis1 does (*Canty and Yildiz, 2020*; *DeSantis et al., 2017*; *Htet et al., 2020*; *Marzo et al., 2020*), one would expect a reduction in gliding velocity when N-DCX is added. However, we found that N-DCX does not affect MT gliding by single-headed dynein (*Figure 5C*), which suggests that N-DCX regulates dynein through interactions with the dynein tail or through binding to dynein's associated subunits. In contrast to N-DCX, we find that DCX decreases the MT gliding velocity slightly (*Figure 5C*). This result implies that DCX also affects dynein-MT interactions through direct MT binding, while N-DCX, which does not bind MTs under the motility buffer conditions (*Figure 5—figure supplement 4B*), does not affect MT gliding powered by the dynein motor domain.

To determine how DCX reduces MT gliding by dynein, we measured how it affects the rate of dynein-MT binding (on-rate) and the time dynein stays MT bound (dwell time) using a TIRF assay with surface-absorbed MTs and GFP-tagged single-headed dynein. Our results show that while DCX does not affect the time dynein stays MT bound (*Figure 5—figure supplement 5A*), it results in a significant reduction in the MT on-rate of dynein (*Figure 5—figure supplement 5B*), which explains the reduced speed of MT gliding by dynein in the presence of DCX. In summary, our data suggest that DCX regulates dynein function via two pathways: through interactions with the dynein motor complex and through its interactions with MTs.

## DCX interferes with the recruitment of a second dynein to DDJ

Previous studies have demonstrated that BicD2 can recruit two dimeric dyneins and that a DDB complex with two dyneins shows higher velocities compared with a DDB complex with one dynein (*Elshenawy et al., 2019*; *Sladewski et al., 2018*; *Urnavicius et al., 2018*). Adaptors such as BicDR1 and Hook3, which predominantly recruit two dyneins, also show increased velocities compared with

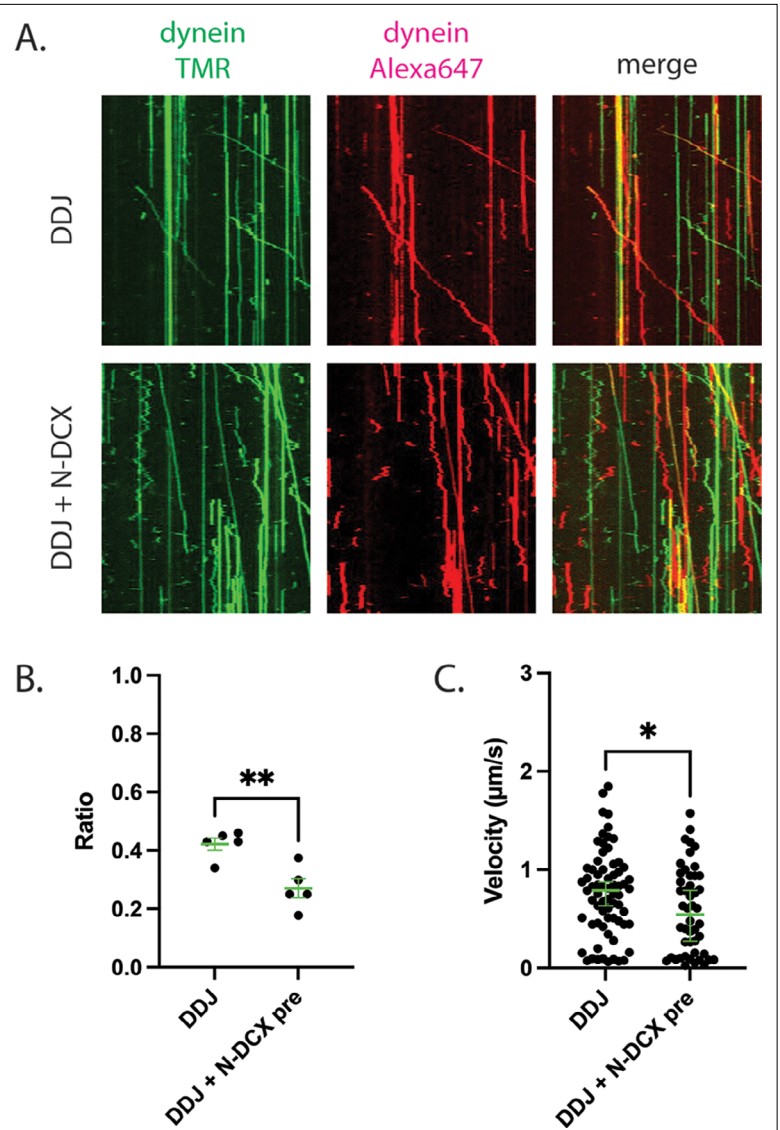

**Figure 6.** DDJ motor complexes associate with two dyneins, and N-DCX negatively affects its velocity by displacing the second dynein. (**A**) Kymograph of DDJ assembled in the absence (top) or presence (bottom) of N-DCX with dynein that were labeled separately with SNAP-TMR and SNAP-Alexa 647. (**B**) The ratio of two-color moving molecules versus the total moving molecules. The green bars represent mean ± SEM. DDJ: 42 ± 2%; DDJ + N-DCX pre: 27 ± 3% (unpaired *t*-test, **p<0.01). The molecules within each field of view were counted to produce a single value (50 μm × 50 μm). (**C**) The velocity of two-color moving molecules. The green bars represent median with 95% CI. DDJ: 0.79 [0.63, 0.87] μm/s; DDJ + N-DCX pre: 0.54 [0.27, 0.79] μm/s (KS test, *p<0.1).

DDR and DDH complexes with only one dynein (*Elshenawy et al., 2019*; *Urnavicius et al., 2018*). We note that the velocity reduction of DDJ in the presence of N-DCX is similar to the velocity reduction when a two-dynein motor assembly loses a dynein motor. Moreover, when we assembled DDJ complexes in the presence of N-DCX, the velocity of DDJ was reduced further (0.41 [0.32, 0.60] μm/s, ****p<0.0001) (*Figure 5C*).

To test the hypothesis that N-DCX displaces the second dynein from a DDJ motor complex with two dyneins, we first determined whether JIP3 permits the recruitment of two dyneins. To do so, we assembled DDJ complexes using equal amounts of dynein labeled with TMR and Alexa 647. In the absence of DCX, we indeed observed moving DDJ complexes with two colocalized colors, demonstrating that DDJ can recruit two dyneins (*Figure 6A*). The fraction of colocalization was 42 ± 2% (mean ± SEM), which is comparable to the colocalization of DDR and DDH complexes with two differently

labeled dyneins (*Elshenawy et al., 2019*; *Urnavicius et al., 2018*). In support of our hypothesis that N-DCX displaces a dynein from a two-dynein motor complex, addition of N-DCX reduced the colocalization fraction to 27 ± 3% (*Figure 6B*), which is close to the reported colocalization of the DDB motor complex (*Elshenawy et al., 2019*). In addition, those few remaining DDJ complexes that contained two colocalized dyneins despite the presence of N-DCX showed a reduced velocity in the presence of N-DCX (*Figure 6C*). In conclusion, similar to DDR and DDH, DDJ predominantly recruits two dyneins, and N-DCX interferes with the binding of the second dynein; N-DCX can still affect the velocity of the DDJ complex with even two dyneins, possibly via disrupting interaction between the tails of the two dyneins (*Elshenawy et al., 2019*).

## Rescuing retrograde transport defects in *Dcx^-/y;Dclk1^-/-* neurons ameliorates neuronal migration defects

One of the characteristics of DCX-linked lissencephaly is a profound defect in cortical neuronal migration. We therefore asked whether the effects of DCX on dynein-based retrograde transport we observe play a role in the migration of cortical neurons during development. If the answer is yes, rescuing the abnormally increased dynein-based retrograde trafficking should mitigate the cortical neuronal migration defects observed in the developing *Dcx^-/y;Dclk1^-/-* mouse brain (*Deuel et al., 2006*; *Koizumi et al., 2006*). Since cortical neuronal migration is relatively normal in the *Dcx^-/y* mouse (*Corbo et al., 2002*), we used a *Dcx^-/y;Dclk1^-/-* mouse, which has a cortical neuronal migration defect as the Dcx-redundant gene Dclk1is knocked out as well (*Deuel et al., 2006*; *Koizumi et al., 2006*). A plasmid expressing GFP and an shRNA that specifically targets DHC (*Tsai et al., 2007*) was microinjected into the lateral ventricle of embryonic day (E)14.5 *Dcx^-/y;Dclk1^-/-* mouse brains and transfected using in utero electroporation. Mouse embryos were then sacrificed on E18.5. As expected, downregulating DHC partially rescued the retention of neuroblasts in the deeper region of the cortex observed in *Dcx^-/y; Dclk1^-/-* mouse brains (*Figure 7A and B*). Based on these results, we wondered whether the dysregulation of dynein is in part due to increased association of JIP3 with dynein in the absence of DCX and whether downregulation of JIP3 expression may also ameliorate neuronal migration defects. To test this possibility, we microinjected plasmids expressing JIP3 shRNA1 and GFP into the lateral ventricle of E14.5 *Dcx^-/y;Dclk1^-/-* embryos and transfected the plasmids into neural progenitors using in utero electroporation. In agreement with our hypothesis, downregulation of JIP3 in *Dcx^-/y;Dclk1^-/-* mouse brain significantly rescued the lamination defect (*Figure 7C and D*). Collectively, our results demonstrate the importance of the regulation of dynein-dependent retrograde trafficking by DCX and JIP3 during neuronal migration.

## Discussion

Previous reports have linked DCX, a causative gene for X-linked lissencephaly that encodes a neural-specific MT-binding protein, to defects in dynein-based functions (*Kaplan and Reiner, 2011*; *Li et al., 2021*; *Tanaka et al., 2004*). How DCX modulates dynein functions has remained unclear. In this study, we demonstrate for the first time that DCX negatively regulates dynein-mediated retrograde trafficking in neuronal axons through its interactions with MTs and through interactions with the dynein motor complex (*Figure 8*). We show that DCX decreases the velocity and processivity of dynein-based cargo transport in vivo and the velocity of dynein-dynactin-JIP3 motor complexes in vitro, and demonstrates that the DCX-based regulation of dynein-driven retrograde transport is important to cortical development. Combined with our previous finding that DCX positively regulates KIF1A-mediated anterograde transport (*Liu et al., 2012*), we conclude that DCX differentially regulates anterograde and retrograde intracellular trafficking in neuronal axons, and, therefore, mediates the transport of critical protein complexes during neuronal growth and development.

## DCX regulates anterograde and retrograde transport through differential effects on MT-associated motors

Our previous work showed that the KIF1A-driven anterograde transport of Vamp2 is significantly decreased in the absence of DCX compared with WT neurons, a result which our in vitro studies indicate is the consequence of a DCX-induced increase in the MT-binding strength of KIF1 in its weak ADP state. However, how DCX regulates KIF5-mediated axonal transport is less clear. Here, our in

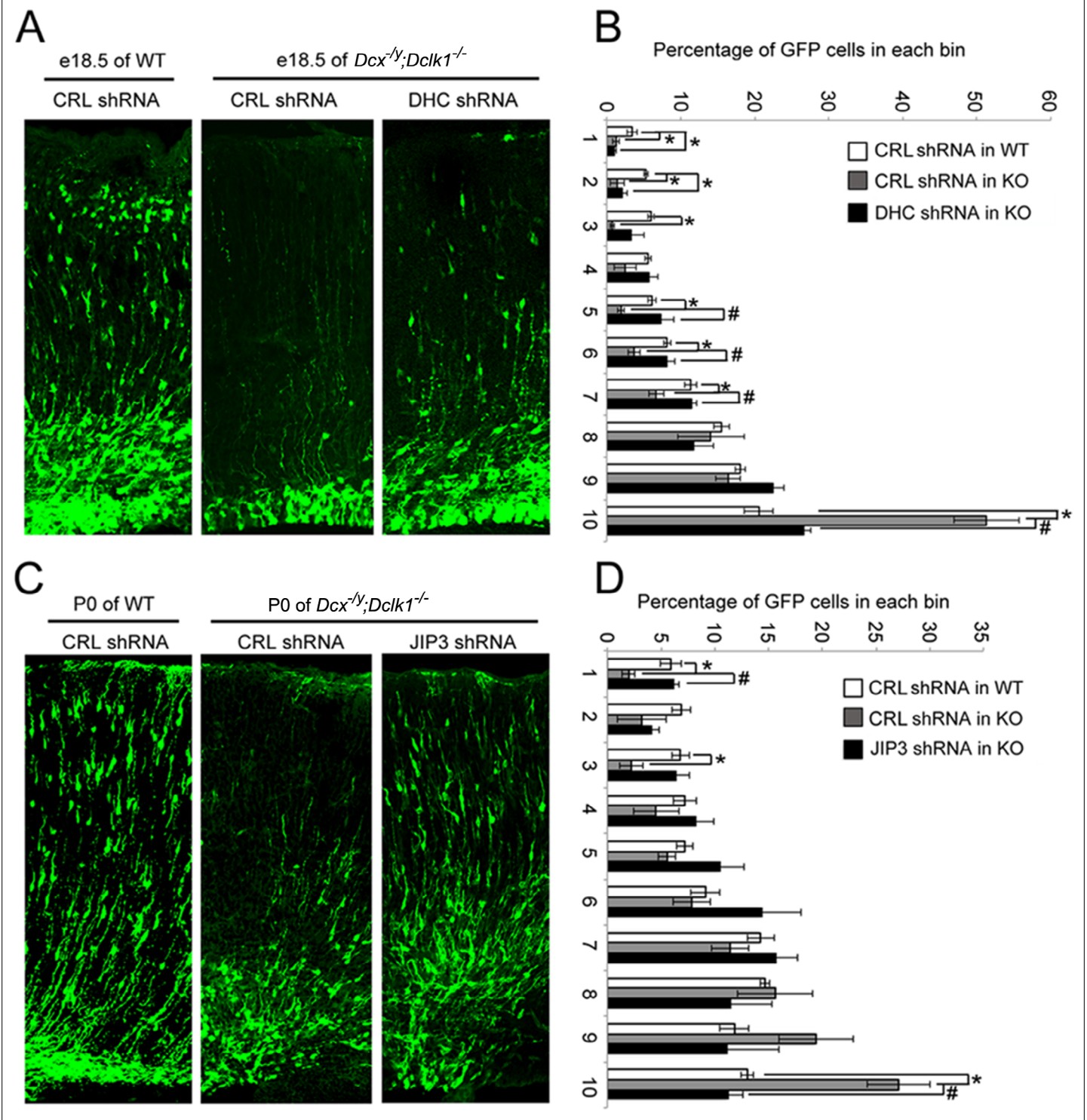

**Figure 7.** Knockdown of DHC or JIP3 in *Dcx$^{-/y}$;Dclk1$^{-/-}$* in mouse cortex partially rescues the defect of pyramidal cell migration. (**A**) GFP-positive neurons were imaged and counted at embryonic day (E)18.5 after electroporation at E14.5 with vectors expressing control (CRL) shRNA (+GFP) or DHC shRNA (+GFP). (**B**) Percent of GFP-positive cells in evenly divided regions of the cortex (1–10) from the pia to the lateral ventricle. Asterisks denote statistically significant p-values (*t*-test, $p<0.05$) between WT with CRL shRNA and *Dc$^{x-/y}$;Dclk1$^{-/-}$* with CRL shRNA. Number sign (#) denotes $p<0.05$ of *t*-test between *Dcx$^{-/y}$;Dclk1$^{-/-}$* with CRL shRNA and *Dcx$^{-/y}$;Dclk1$^{-/-}$* with DHC shRNA. The data represent the mean ± SEM of three different brains in each condition. (**C**) GFP-positive neurons were imaged and counted at P0 after electroporation at E14.5 with vectors expressing control (CRL) shRNA (+GFP) or JIP3 shRNA (+GFP). (**D**) Percent of GFP-positive cells in evenly divided regions of the cortex (1–10) from the pia to the lateral ventricle of different mouse brains. * denotes $p<0.05$ of *t*-test between WT with CRL shRNA and *Dcx$^{-/y}$;Dclk1$^{-/-}$* with CRL shRNA. Number sign (#) denotes $p<0.05$ of *t*-test between *Dcx$^{-/y}$;Dclk1$^{-/-}$* with CRL shRNA and *Dcx$^{-/y}$;Dclk1$^{-/-}$* with JIP3 shRNA. The data represent the mean ± SEM of three individual brains in each condition.

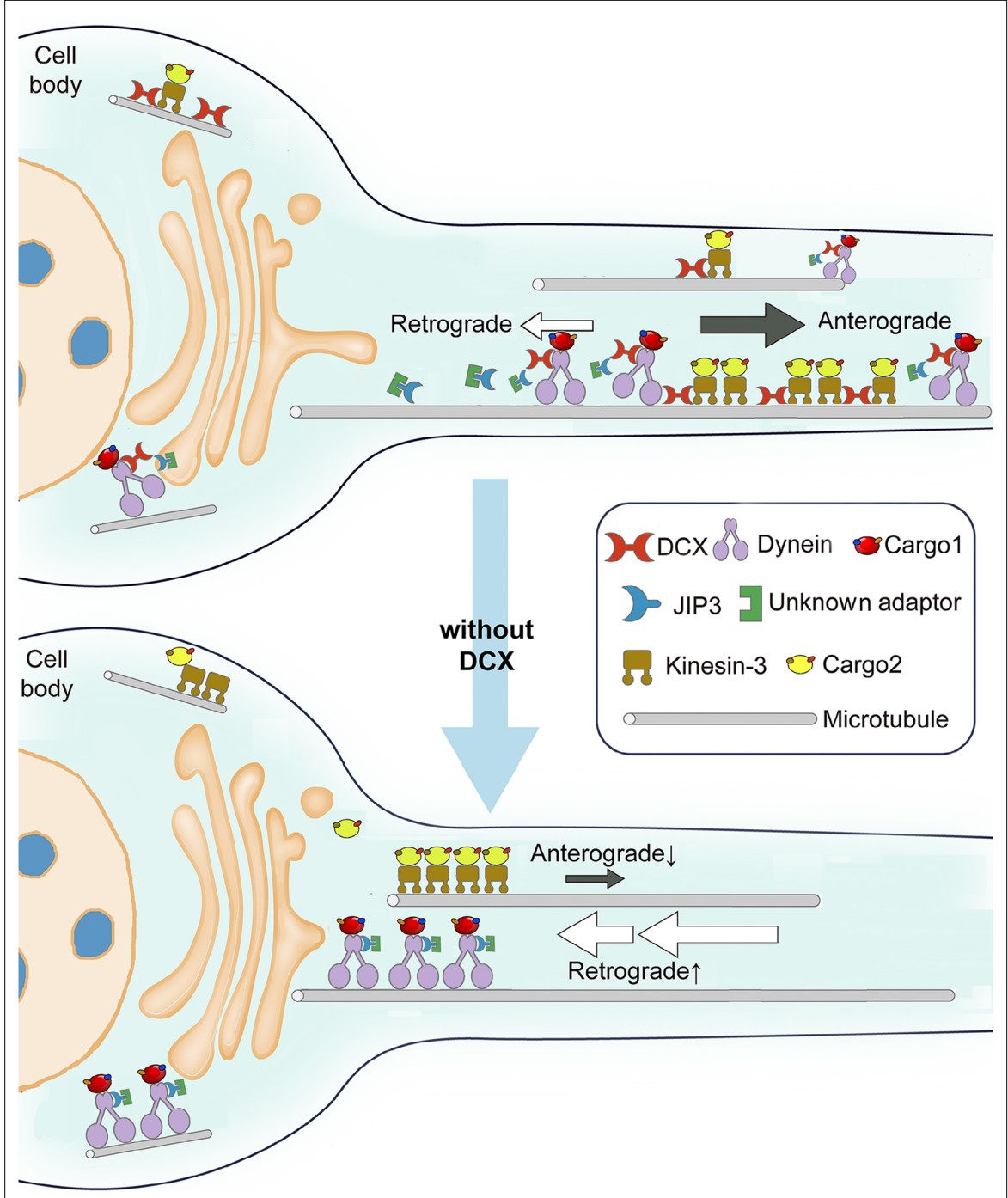

**Figure 8.** Schematic diagram shows the regulation of dynein-mediated retrograde transport by doublecortin (DCX). Cargo-bound dynein motor complex drives retrograde transport from plus end of microtubules (MTs) (distal axon) to minus end of MTs (cell body). In WT neurons, DCX association with kinesin-3 helps kinesin-3-mediated anterograde transports (*Liu et al., 2012*). DCX decreases dynein-MT interactions (represented by tilted dynein complex along MT). DCX and JIP3 competitively associate with dynein. When DCX binds dynein, very few JIP3 proteins associate with dynein, the retrograde transport is normal. In DCX KO neurons (without DCX), kinesin-3-mediated anterograde transports are decreased without DCX (*Liu et al., 2012*). Meanwhile, more JIP3 molecules bind dynein, which also associates with MT stronger without DCX. The dynein-mediated retrograde transport is faster. The balance between anterograde transport and retrograde transport is broken without DCX.

vitro data show that DCX reduces the velocity of KIF5B, consistent with a previous report (*Monroy et al., 2020*). Yet, we find that the anterograde transport of TrkB, which is regulated by KIF5 (*Arimura et al., 2009*; *Huang et al., 2011*; *Sun et al., 2017*), remains unchanged in DCX knockout neurons. The anterograde transport of mitochondria, which is mainly mediated by KIF5 (*Cai et al., 2005*; *Iqbal and Hood, 2014*; *Lawrence et al., 2016*; *Morcillo et al., 2021*; *Qin et al., 2020*; *Zorgniotti et al., 2021*), is also unaffected by DCX deficiency, both in terms of percentage of mobile mitochondria and run length (*Liu et al., 2012*).

This begs the question: why have no significant changes in KIF5-mediated anterograde transport in DCX-deficient neurons been observed when the in vitro data clearly suggest that DCX negatively regulates KIF5 motility? In the case of TrkB transport, one possibility is that the effects of DCX deficiency on KIF1A and KIF5B counteract each other. A recent study demonstrated that KIF5 and KIF1 cooperate in the anterograde transport of TrkB from the secretory compartment into and along the axon (*Zahavi et al., 2021*). Therefore, DCX's promoting effect on KIF1A might counteract DCX's inhibitory effect on KIF5, the combined effects canceling each other out, so that no differences in anterograde transport of TrkB occur with DCX deficiency. Another possibility is that changes in JIP3 binding to KIF5 in the absence of DCX counteract the decreased negative regulation of KIF5. JIP3 is involved in the anterograde transport of TrkB by KIF5 (*Arimoto et al., 2011*; *Huang et al., 2011*). We found that the interaction of JIP3 with dynein was increased in DCX-deficient neurons while the total JIP3 levels were not changed compared with WT neurons (*Li et al., 2021*). It is then possible that less JIP3 associates with KIF5 in the absence of DCX, thereby decreasing KIF5 motility and counteracting the increased velocity of KIF5B. A third possibility is based on the suggestion that kinesin and dynein engage in a 'tug-of-war' when attached to the same cargo (*Belyy et al., 2016*; *Gennerich and Schild, 2006*; *Rezaul et al., 2016*): DCX deficiency may shift the balance of forces between dynein and kinesin motors so that KIF5B's increased velocity is counteracted by enhanced dynein motility in the opposite direction.

These possibilities are not mutually exclusive and illustrate the complexity of the network of factors regulating intracellular transport, in general, and the complexity of DCX regulation of axonal transport, in particular. DCX regulates axonal transport through multiple related pathways, and the net impact of DCX deficiency on vesicular transport is therefore the sum of alterations in these interrelated pathways. What is clear is that DCX affects MT-associated motor proteins differentially to regulate both anterograde and retrograde cargo transport. Our data here show that DCX accomplishes this in part through direct interactions with at least two MT motors, the KIF1A and the DDJ complex.

## DCX negatively regulates dynein motion through interactions with both dynein and MTs

We show that DCX inhibits dynein-mediated transport directly, both through its interactions with the dynein motor complex and through its binding to MTs. We demonstrate that DCX interacts with dynein through its N-terminal domain (*Figure 2E*) and that the C-terminal domain decreases binding of DCX to dynein, suggesting that DCX binds dynein in a 'closed' conformation (facilitated through intramolecular interactions between the N-terminal and C-terminal domains) in which the C-terminal domain is incapable of interfering with DCX-dynein binding. This could suggest that DCX binds to dynein as an autoinhibited monomer.

The binding of DCX to MTs also impacts retrograde transport based on our observation that the pathogenic DCX mutations A71S and T203R fail to restore dynein-mediated retrograde trafficking. These mutations decrease the cooperative binding of DCX to MTs but do not affect the interactions of DCX with dynein. We conclude that MT-bound DCX impairs retrograde transport by reducing the initial MT binding or the rebinding of the tethered dynein head during forward stepping. However, given that DCX also inhibits dynein transport by binding to the DDJ complex (an interaction that is unaffected by these mutations), we might have expected to observe at least a partial rescue of the dynein transport phenotype with A71S and T203R expression. A possible explanation is that the cooperative binding of DCX to MTs in vivo increases the local DCX concentration, which in turn increases the chances that a DCX molecule associates with a nearby DDJ complex, causing the second dynein to dissociate and thereby reducing the velocity of the DDJ complex. As the A71S and T203R mutants cannot accumulate locally on MTs, they may have a reduced chance of binding to the DDJ complex and thus show a significantly reduced inhibitory effect on retrograde transport.

DCX binds MTs in a dimerized form or as oligomers of higher order, and it has been shown that this DCX self-association is an MT-dependent phenomenon (*Moores et al., 2006*; *Rafiei et al., 2022*). Therefore, the overall conformation of DCX changes when DCX associates with MTs. Studies also showed that DCX dynamically associates with MTs mainly through its N-terminal R1 MT-binding domain (*Moslehi et al., 2017*; *Rafiei et al., 2022*). However, DCX's C-terminal region still plays a critical role in the association of DCX with MTs despite C-DCX's inability to directly bind to MTs (*Rafiei et al., 2022*). The C-terminus of DCX facilitates the binding of neighboring DCX molecules to MTs through its interactions with the N-terminal domain of the neighboring DCX (*Rafiei et al., 2022*). This suggests that the C- and N-terminal domains of DCX have an intrinsic affinity for each other (possibly resulting in a 'closed' conformation as discussed above) and that MT binding leads to a shift from intramolecular interactions between the N-terminal and C-terminal domains to intermolecular interactions between the domains of neighboring MT-bound DCX molecules. The proposed function for the C-terminus in DCX-MT binding is consistent with our MT pull-down results, which demonstrate that significantly more DCX is bound to MTs in the presence of C-DCX (*Figure 3B*).

We hypothesize that the association of DCX with MTs and the dynein complex is a dynamic and of competitive nature. When more DCX associates with MT, less will bind to the dynein complex and vice versa. This dynamic process is likely regulated directly by the interactions of the C-terminal and N-terminal domains or indirectly through phosphorylation of residues in the C-terminal domain. Since the C-terminal domain has several phosphorylation sites (*Graham et al., 2004*; *Jin et al., 2010*; *Shmueli et al., 2006*; *Slepak et al., 2012*; *Tanaka et al., 2004*), it will be interesting in future studies to determine whether phosphorylation of residues in the C-terminal domain regulates the intramolecular interactions of the C- and N-terminal domain and with it DCX's 'closed' state. In doing so, phosphorylation of the C-terminal domain could regulate whether DCX preferentially binds MTs or the DDJ complex.

## DCX regulates dynein-mediated retrograde transport through JIP3

DCX's association with dynein also alters the composition of the dynein motor complex. We found that the presence/absence of DCX most strongly altered the amount of the signaling adaptor protein, JIP3, that immunoprecipitated with the dynein motor complex (*Li et al., 2021*). In this study, we find that DCX and JIP3 competitively associate with dynein and that a DCX-induced reduction in the association of JIP3 with dynein results in diminished dynein-mediated retrograde transport. Thus, when DCX is absent, more dynein motors associate with MTs, and more JIP3 associates with dynein – events that greatly promote retrograde trafficking of TrkB.

JIP3 belongs to the JIP family of proteins, and all mammalian JIP proteins are expressed in the brain (*Dickens et al., 1997*; *Ito et al., 1999*; *Kelkar et al., 2000*; *Kelkar et al., 2005*; *Yasuda et al., 1999*). JIP3 is an adaptor that regulates both anterograde and retrograde transport through its binding to kinesin and dynein, respectively (*Celestino et al., 2022*). Sunday Driver, the JIP3 homolog in *Drosophila*, directly binds to kinesin-1 (*Bowman et al., 2000*; *Byrd et al., 2001*; *Sun et al., 2011*), and UNC16, the JIP3 homolog in *C. elegans*, interacts with both kinesin-1 and dynein (*Byrd et al., 2001*). JIP3 colocalizes with the dynein-dynactin motor complex and serves as an adaptor protein for dynein-mediated retrograde transport of active JNK and lysosomes (*Cavalli et al., 2005*; *Drerup et al., 2013*). We found that the interaction of JIP3 with dynein was increased in DCX-deficient neurons, although the total JIP3 levels were not changed compared with WT neurons (*Li et al., 2021*). These findings are supported by our in vitro data, which demonstrate that while JIP3 predominantly recruits two dyneins per DDJ motor complex in the absence of DCX, it mostly recruits only one dynein in the presence of DCX. As JIP3 is also involved in the anterograde transport of TrkB by KIF5 (*Arimoto et al., 2011*; *Huang et al., 2011*), it will be interesting to determine whether JIP3 association with KIF5 is affected by DCX.

## DCX regulates the assembly and motility of the dynein-dynactin-JIP3 motor complex

We have revealed that DCX competes with the binding of the second dynein to DDJ, resulting in reduced velocities of the moving motor complexes by reconstituting for the first time the in vitro motility of DDJ motor complexes using two-color single-molecule colocalization studies. While numerous dynein-dynactin-adaptor complexes such as DDB, DDR, and DDH have been extensively

studied in vitro using single-molecule TIRF microscopy (*Christensen et al., 2021*; *McClintock et al., 2018*; *McKenney et al., 2014*; *Sladewski et al., 2018*; *Urnavicius et al., 2018*), it was the consensus that the predicted ~180 amino acids α-helical coiled-coil region in JIP3 is too short to be capable of forming a tripartite complex with dynein and dynactin (*Chaaban and Carter, 2022*; *Lee et al., 2020*; *Reck-Peterson et al., 2018*). Surprisingly, however, we found that a truncated JIP3 construct containing the N-terminal coiled-coil and the predicted adjacent intrinsically disordered domain (amino acids 1–240) can form an active DDJ complex with two dyneins. N-DCX reduced the velocity of DDJ complexes with two dyneins from ~0.8 µm/s to ~0.4 µm/s as a result of the dissociation of the second dynein. Moreover, we found that N-DCX, which does not bind MTs under our assay condition, does not impact the MT-gliding activity by a tail-truncated single-head dynein (a recombinant construct that contains the motor domain and the linker), while FL-DCX, which decorates MTs well, reduces the MT-gliding velocity slightly. These results suggest that N-DCX's inhibitory effect on DDJ is through its interactions with dynein's tail domain or dynein's associated subunits, but not through interactions with dynein's motor domain or the MTs. On the other hand, FL-DCX decreases dynein's MT-gliding velocity slightly, and our single-molecule landing-rate assay showed that while DCX does not alter the time of dynein spends bound to MTs, it reduces the MT on-rate of dynein. Thus, DCX affects the composition of the DDJ complex as well as dynein's interactions with MTs.

The fact that DDJ complexes are active and associate with two dyneins implies that either the predicted disordered region in JIP3 following the first α-helix forms an α-helical structure (possibly when JIP3 interacts with dynactin and the dynein tail) or the minimal length for the coiled-coil region of the adaptor does not need to span the full dynactin shoulder. This result contrasts with a previous report that demonstrated that while a short Hook3 construct could form a stable complex with dynein and dynactin, the resulting DDH complex was incapable of moving along MTs (*Schroeder and Vale, 2016*). It is possible that Hook3 and JIP3 interact differently with dynactin and dynein, which could result in a different degree of activation. Indeed, cryo-EM studies not only show that different adaptors bind to dynactin and dynein differently (*Urnavicius et al., 2018*) but also that two Hook3 adaptors (or two BicD adaptors) can bind dynactin simultaneously to recruit two dyneins (*Chaaban and Carter, 2022*), while a recent *bioRxiv* paper shows that only one JIP3 molecule is sufficient to recruit two dyneins to dynactin (*Singh et al., 2022*). These studies clearly demonstrate that JIP3 interacts differently with dynactin and the two dyneins to form the DDJ motor complex, and suggest that DCX displaces the second dynein from the DDJ complex without disrupting the interaction of JIP3 with dynactin. DCX therefore competes with the interaction of dynein with JIP3 but not with the interaction of JIP3 and dynactin.

Our findings collectively suggest that cargo adaptors fine-tune dynein's activity by utilizing different interactions with dynein and dynactin. While we also found that a longer JIP3 construct (aa 1–548) formed an active complex with dynein and dynactin (data not shown), this construct was prone to aggregation. Of note, full-length BicD2 has been shown to be autoinhibited by its third coiled-coil domain (*Hoogenraad et al., 2001*; *McClintock et al., 2018*). It is therefore possible that our longer JIP3 construct is more prone to be autoinhibited. To circumvent this problem, we used the shorter JIP3 construct for our studies. Collectively, our in vitro reconstitution studies with purified proteins agree with our in vivo observations that DCX downregulates dynein's activity, and that the C-terminus of DCX autoinhibits DCX's interaction with dynein.

## How does disinhibition of dynein by DCX-based loss of function lead to defects in early neuronal development?

Both loss-of-function mutations in DCX, dynein, and its cofactors cause cortical malformations (*Feng and Walsh, 2004*; *Pawlisz et al., 2008*; *Poirier et al., 2013*; *Reiner et al., 1993*; *Sasaki et al., 2005*; *Youn et al., 2009*), and JIP3 knockout in mouse results in similar phenotypes as seen in *DCX -/y*; *Dclk1-/-* mice (*Deuel et al., 2006*; *Fu et al., 2013*; *Koizumi et al., 2006*) with disrupted formation of the telencephalon and the agenesis of the telencephalic commissures, possibly through impaired vesicle transport and defects in axon guidance (*Ha et al., 2005*; *Kelkar et al., 2003*). Furthermore, previous studies showed that JIP3 regulates axon branching through GSK3β-signaling pathway by phosphorylation of DCX at Ser327, which is located at C-terminal S/P-rich region of DCX (*Bilimoria et al., 2010*). The phosphorylation of DCX by the JNK pathway is important for neuronal migration

(*Gdalyahu et al., 2004*). Thus, loss-of-function mechanisms in cytoskeleton motor proteins are well-accepted as causative in neural developmental disorders.

However, in this study we show that dynein function is abnormally increased in DCX null neurons. Furthermore, we demonstrate that diminishing dynein activity by either knocking down DHC or by decreasing the amount of JIP3 can partially rescue the defects in $Dcx^{-/y};Dclk^{-/-}$ mouse cortex. The fact that the rescue is regional and affects deeper regions of the cortex near the ventricular and subventricular zone implies that early defects in neural progenitor or neuroblast biology may be preferentially affected by increased dynein activity in mice with lacking DCX. When dynein activity is abnormally and globally increased, precise spatial regulation of motor function is lost in neural progenitors and neurons, which may have effects on cell-biological functions mediated by dynein, including progenitor cell division, nucleokinesis, and polarized transport of signaling molecules (*Roberts et al., 2013*; *Tsai et al., 2010*; *Vale, 2003*). Dynein and KIF1A regulate apically directed or basally directed nuclear movement, respectively, of radial glial progenitor cells (*Tsai et al., 2010*). It is suggested that DCX mediates KIF1A's effect on basally directed nuclear movement through the BDNF pathway (*Carabalona et al., 2016*), while influencing dynein's role in apically directed nuclear movement through regulating the perinuclear MT structure (*Tanaka et al., 2004*). Based on our results in this study, DCX might also regulate nuclear migration through influencing the balance of KIF1A/dynein-mediated anterograde/retrograde transport via its regulation of JIP3 association with the two motors.

Furthermore, loss of dynein inhibition may have direct effects on important signal transduction pathways from distal neuronal processes, including neurotrophin (BDNF) (*Bhattacharyya et al., 2002*) and mitogen-activated protein kinase signaling via JNK (*Rishal and Fainzilber, 2014*). JIP3 is known to bind dynein upon JNK activation and is, thus, an important mediator of the mitogen-activated protein kinases (*Drerup et al., 2013*). While JIP3's effects on BDNF signaling have not been appreciated previously, JIP3 enhanced retrograde transport of the canonical BDNF cargo, TrkB, is consistent with the known cross-talk between neurotrophic and mitogen-activated protein kinase signaling (*Huang et al., 2011*). Alternatively, JIP3 may be a more general dynein cofactor for mediating retrograde trafficking. While DCX's effects on dynein and JIP3 occur predominantly in development, the activity of other DCX-family proteins, which are expressed in mature neurons (*Reiner et al., 2006*), has important consequences for understanding neuron-specific signaling that extend beyond development to degeneration, injury, and repair. Further studies are needed to understand the coordinate regulation of motors by DCX.

## Materials and methods

**Key resources table**

| Reagent type (species) or resource | Designation | Source or reference | Identifiers | Additional information |
|---|---|---|---|---|
| Genetic reagent (*Mus musculus*) | $Dcx^{-/y}$, 129/SvJ | *Deuel et al., 2006* | | |
| Genetic reagent (*M. musculus*) | $Dcx^{-/y};Dclk1^{-/-}$, 129/SvJ | *Deuel et al., 2006* | | |
| Cell line (*Homo sapiens*) | HEK293 | ATCC | CRL-1573 | The identity of HEK293 was authenticated from ATCC upon purchase and tested negative for mycoplasma contamination |
| Strain, strain background | BL21-CodonPlus (DE3)-RIPL competent cells | Agilent | #230280 | |
| Strain, strain background | DH10Bac competent cells | Gibco | #10361012 | |
| Antibody | Anti-DCX (rabbit polyclonal) | Abcam | ab18723 | (1:1000 dilution) |
| Antibody | Anti-DHC (mouse monoclonal) | Abcam | ab6305 | (1:250 dilution) |
| Antibody | Anti-DIC (mouse monoclonal) | Abcam | ab23905 | (1:1000 dilution) |
| Antibody | Anti-Tubulin (rat monoclonal) | Abcam | ab6161 | (1 µg/ml dilution) |
| Antibody | Anti-JIP3 (rabbit polyclonal) | Abcam | ab196761 | (1:1000 dilution) |

*Continued on next page*

*Continued*

| Reagent type (species) or resource | Designation | Source or reference | Identifiers | Additional information |
|---|---|---|---|---|
| Antibody | Anti-HA (mouse monoclonal) | EMD Millipore | 05-904 | (1 µg/ml dilution) |
| Antibody | Anti-GFP (mouse monoclonal) | YenZym | https://www.yenzym.com/default.html | (0.1 mg/ml) |
| Antibody | Alexa 488-conjugated goat anti-rabbit IgG (H+L) (goat polyclonal) | Thermo Fisher Scientific | A11036 | (1:10,000 dilution) |
| Antibody | Alexa 568-conjugated goat anti-rabbit IgG (H+L) (goat polyclonal) | Thermo Fisher Scientific | A-11036 | (1:10,000 dilution) |
| Antibody | Alexa 488-conjugated goat anti-mouse IgG (H+L) (goat polyclonal) | Thermo Fisher Scientific | A32723 | (1:10,000 dilution) |
| Antibody | Alexa 568-conjugated goat anti-mouse IgG (H+L) (goat polyclonal) | Thermo Fisher Scientific | A-11031 | (1:10,000 dilution) |
| Recombinant DNA reagent | 6His-PreScission-DCX-EGFP-StrepII (plasmid) | Addgene | #83918 | |
| Recombinant DNA reagent | kif5b(1-560)-EGFP-6His (plasmid) | Addgene | #15219 | |
| Recombinant DNA reagent | Sfp-6His (plasmid) | Addgene | #75015 | |
| Recombinant DNA reagent | Construct pGEX-4T-1 encoding JIP3 (plasmid) | Gift from Dr. Valeria Cavalli; *Sun et al., 2011* | | |
| Recombinant DNA reagent | Construct pCDNA3 encoding JIP3 (plasmid) | Gift from Dr. Roger Davis; *Kelkar et al., 2000* | | |
| Recombinant DNA reagent | pSNAP-tag(T7)2 (plasmid) | NEB | NEB #N9181S | |
| Recombinant DNA reagent | pFastBac plasmid with codon-optimized full-length human dynein (plasmid) | A gift from the Carter lab; *Schlager et al., 2014* | | |
| Recombinant DNA reagent | The pFastBac plasmid encoding tail-truncated human dynein (amino acids 1320–4646) (plasmid) | Gift from the Reck-Peterson Lab; *Htet et al., 2020* | | |
| Recombinant DNA reagent | Construct expressing IC-1B (plasmid) | Gift from Dr. Kevin Pfister; *Ha et al., 2008* | | |
| Recombinant DNA reagent | Construct expression TrkB-RFP (plasmid) | Gift from Dr. Xiaowei Zhuang | | |
| Recombinant DNA reagent | Plasmid pBA (plasmid) | Gift from Dr. Gary Banker; *Jacobson et al., 2006* | | |
| Recombinant DNA reagent | The pBA plasmid encoding HA-tagged N-DCX mutant (plasmid) | This paper | Available upon request | 'Materials and methods' |
| Recombinant DNA reagent | The pBA plasmid encoding HA-tagged C-DCX mutant (plasmid) | This paper | Available upon request | 'Materials and methods' |
| Recombinant DNA reagent | The pBA plasmid encoding HA-tagged DCX mutant T203R (plasmid) | This paper | Available upon request | 'Materials and methods' |

*Continued on next page*

*Continued*

| Reagent type (species) or resource | Designation | Source or reference | Identifiers | Additional information |
|---|---|---|---|---|
| Recombinant DNA reagent | The pBA plasmid encoding HA-tagged DCX mutant A71S (plasmid) | This paper | Available upon request | 'Materials and methods' |
| Recombinant DNA reagent | pSilencer-GFP (plasmid) | Gift from Dr. Shirin Bonni; *Sarker et al., 2005* | | |
| Recombinant DNA reagent | pSilencer-GFP plasmid encoding JIP3 shRNA (plasmid) | This paper; *Li et al., 2021* | | 'Materials and methods' |
| Chemical compound, drug | S.O.C. medium | Gibco | #15544034 | |
| Chemical compound, drug | B-PER complete | Thermo Scientific | #89821 | |
| Chemical compound, drug | QIAGEN buffer | QIAGEN | #27104 | |
| Chemical compound, drug | FuGene HD transfection reagent | Promega | E2311 | |
| Chemical compound, drug | Protease inhibitor cocktail tablet | Roche | #11836170001 | |
| Chemical compound, drug | IgG Sepharose 6 Fast Flow beads | Cytiva | #17096901 | |
| Chemical compound, drug | SNAP-Cell TMR | New England Biolabs | #S9105S | |
| Chemical compound, drug | TEV protease | New England Biolabs | #P8112S | |
| Chemical compound, drug | Tubulin | Cytoskeleton | #TL590M-A | |
| Chemical compound, drug | Zeba 7 kDa unit | Thermo Scientific | #89882 | |
| Chemical compound, drug | Dynactin | A gift from the laboratory of Andrew Carter, MRC | | |
| Chemical compound, drug | Lipofectamine 2000 | Thermo Fisher Scientific | #11668019 | |
| Chemical compound, drug | Polyornithine | Sigma-Aldrich | #P4957-50ML | |
| Chemical compound, drug | B27 | Thermo Fisher Scientific | #17504044 | |
| Chemical compound, drug | bFGF | Thermo Fisher Scientific | #PHG0024 | |
| Chemical compound, drug | Papain dissociation system | Worthington Biochemical Corporation | LK003150 | |
| Software, algorithm | MetaMorph Premier | MetaMorph | https://www.metamorphsoftware.com/ | |
| Other | Six-well plates | Corning | #3516 | Used in neuronal culture |
| Other | Glass coverslips (for 24-well) | Warner Instruments | Cat# 64-0712 | Used in neuronal culture |
| Other | Glass coverslips (for 6-well plate) | Warner Instruments | Cat# 64-0715 | Used in neuronal culture |

## Antibodies and reagents

Cell culture reagents were purchased from Life Technologies (Grand Island, NY). HEK-293 cells were purchased from ATCC (#CRL-1573). The authentication of this cell lines was provided by ATCC, and

the cell line was tested negative for mycoplasma contamination. Antibodies to DCX (ab18723), DHC (ab6305), DIC (ab23905), JIP3 (ab196761), and Tubulin (ab6161) were purchased from Abcam (Cambridge, MA). Antibody to HA was from EMD Millipore (Billerica, MA). Construct expressing IC-1B was generously provided by Dr. Kevin Pfister (UVA). Construct expression JIP3 was generously provided by Dr. Roger Davis (UMASS MED). Construct expression TrkB-RFP was generously provided by Dr. Xiaowei Zhuang (Harvard University). All other reagents were purchased from Sigma-Aldrich (St. Louis, MO).

## Mammalian expression and RNA interference constructs

DNA sequences for HA-tagged N-DCX (1–270 N-terminal amino acids) and C-DCX (271–361 amino acids) were synthesized by PCR using construct expressing FL-DCX (*Liu et al., 2012*) as template, and then cloned into plasmid pBA (*Jacobson et al., 2006*). HA-tagged DCX mutant T203R were created using QuikChange Site-Directed Mutagenesis kit (Stratagene). HA-tagged DCX mutant A71S was synthesized commercially (Genewiz) and subcloned into plasmid pBA. All RNAi control or target sequences (hp) were cloned into the pSilencer 1.0-U6 plasmid. The complementary RNAi oligos were annealed and ligated into pSilencer-GFP (gift from Shirin Bonni) (*Sarker et al., 2005*).

## Animals and primary cortical neuron cultures

All animal procedures were approved by the Committee on the Ethics of Animal Experiments of Wenzhou Medical University (#wydw2019-0723). P0 cortices were dissected and dissociated using the Worthington papain dissociation system (Worthington Biochemical Corp., Lakewood, NJ). Neurons were plated on poly-L-ornithine solution-coated coverglasses in neuronal culture medium (Neurobasal medium plus B27, glutamine, FGF [10 ug/ml] and Pen/Strep) until experiments.

## Time-lapse imaging

Cultured cortical neurons were transfected with different constructs on DIV6 using Lipofectamine 2000 according to the manufacturer's instruction. Images were acquired on an inverted epifluorescence microscope (IX-81, Olympus America Inc, Melville, NY) equipped with high numerical aperture lenses (Apo 603 NA 1.45, Olympus) and a stage top incubator (Tokaihit, Japan) maintained at 37°C at a rate of one capture per 3 s. Fluorescence excitation was carried out using solid-state lasers (Melles Griot, Carlsbad, CA) emitting at 488 nm (for green) and 561 nm (for red) fluorophores. Emission was collected through appropriate emission band-pass filters obtained from Chroma Technologies Corp. (Brattleboro, VT). Images were acquired with a 12-bit cooled CCD ORCA-ER (Hamamatsu Photonics) with a resolution of resolution of 1280×1024 pixel size of 6.45 mm². The camera, lasers, and shutters were all controlled using Slidebook 5 (Intelligent Imaging Innovations, Denver, CO). For all calculations and measurements of vesicle movement, only bright vesicles located in the proximal region of axons (~100 μm away from cell body) are analyzed. A vesicle is counted as mobile only if the displacement is at least 5 μm. A vesicle is counted as stationary if it moves less than 5 μm. To calculate the run length and velocity, vesicles were analyzed only if the net run length is at least 5 μm in retrograde direction. The velocity is calculated as the length of a continuous retrograde movement divided by the length of the time. Those stationary vesicles are not counted for velocity. Analysis of time-lapse imaging was performed with MetaMorph for tracking and the ImageJ Manual Tracking plugin as described (http://rsbweb.nih.gov/ij/plugins/track/track.html).

## Pull-down assay and mass-spectrometry procedure and analysis

HA or HA-tagged DCX proteins were immobilized on anti-HA agarose beads and subsequently mixed with protein lysates from E18 mouse brains and incubated with rotation for 16 hr at 4°C to pull down associating proteins. The beads were washed four times. The beads were then incubated with DL-dithiothreitol (DTT) solution (final concentration of 10 mmol/l) and reduced in a 56°C water bath for 1 hr. Iodoacetamide (IAA) solution was added (final concentration of 50 mmol/l) and protected from light for 40 min. The proteins were digested with trypsin overnight at 37°C. After digestion, the peptides were desalted using a desalting column, and the solvent was evaporated in a vacuum centrifuge at 45°C. The peptides were dissolved in sample solution (0.1% formic acid [FA] in water) and ready for mass-spectrometry analysis. Samples were loaded onto Nanocolumn (100 μm × 10 cm) packed with a reversed-phase ReproSil-Pur C18-AQ resin (3 μm, 120 Å, Dr. Maisch GmbH, Germany).

The mobile phases consisted of A (0.1% FA in water) and B (acetonitrile [ACN]). Total flow rate is 600 nl/min using a nanoflow liquid chromatograph (Easy-nLC1000, Thermo Fisher Scientific, USA). LC linear gradient: from 4% to 8% B for 2 min, from 8% to 28% B for 43 min, from 28% to 40% B for 10 min, from 40% to 95% B for 1 min, and from 95% to 95% B for 10 min. Eluted peptides were introduced into the mass spectrometer (Q Exactive Hybrid Quadrupole-Orbitrap Mass Spectrometer, Thermo Fisher Scientific). The spray voltage was set at 2.2 kV and the heated capillary at 270°C. The machine was operated with MS resolution at 70,000 (400 m/z survey scan), MS precursor m/z range: 300.0–1800.0. The raw MS files were analyzed and searched against protein database based on the species of the samples using MaxQuant (1.6.2.10). The parameters were set as follows: the protein modifications were carbamidomethylation (C) (fixed), oxidation (M) (variable), acetyl (protein N-term) (variable); the enzyme specificity was set to trypsin; the maximum missed cleavages were set to 2; the precursor ion mass tolerance was set to 20 ppm, and MS/MS tolerance was 20 ppm. Only high-confident identified peptides were chosen for downstream protein identification analysis. RIPA Lysis and Extraction Buffer, Pierce BCA Protein Assay Kit were purchased from Thermo Fisher Science. DTT, IAA, FA, ACN were purchased from Sigma (St. Louis, MO), and trypsin from bovine pancreas was purchased from Promega (Madison, WI). Ultrapure water was prepared from a Millipore purification system. An Ultimate 3000 system was coupled with a Q Exactive Hybrid Quadrupole-Orbitrap Mass Spectrometer (Thermo Fisher Scientific) with an ESI nanospray source.

## Microtubule-binding assay

Mouse brains are dissected and flash-frozen and kept at –80°C until experiment. Flash-frozen mouse brains are pulverized with a mortar and pestle and added to cold lysis buffer (0.01% Triton X100, 1× proteinase and phosphatase inhibitor cocktail, 1 mM GTP in 1× BRB80 buffer) and left on ice for 20 min. Proteins were collected in the supernatant after centrifugation for 20 min at 15,000 rpm. Tubulin (Cytoskeleton, Inc) was diluted to 10 mg/ml in lysis buffer and incubated at 37°C for 30 min, 100 µM taxol was added afterward. Equal amounts of proteins from different mouse brains were warmed up to 37°C and incubated with polymerized MTs at 37°C for 1 hr. Samples are centrifuged at 100,000 × $g$ at 37°C for 40 min. Supernatants were saved and pellets resuspended in lysis buffer of the same volume of supernatant.

## In utero electroporation

In utero electroporation-mediated gene transfer was performed as previously described (*Saito and Nakatsuji, 2001*; *Tabata and Nakajima, 2001*). Briefly, E14.5 pregnant mice were anesthetized with ketamine/xylazine (100/10 mg/kg) and their uterine horn exposed. DNA plasmid (2–5 mg/ml) was injected via a pulled glass pipette into the lateral ventricle of each embryo, followed by electrodes placed on each side of the head parallel to the sagittal plane. Electrical current (five 50 ms pulses of 41 V with 950 ms intervals) was used to drive the plasmid DNA into lateral cortical areas. After sacrifice, mice were screened through visualizing of GFP expression using a stereo fluorescence microscope. GFP-expressing mouse brains were dissected out and fixed in 3.7% paraformaldehyde for 3 hr. Samples were then transferred to PBS buffer with 30% sucrose and left at 4°C overnight. The mouse brains were sectioned at 20 µm using a Microtome (MICROM HM525).

## Western analysis

Standard Western blot analysis was performed using antibodies as detailed above. The dual-channel signal detection LI-COR system from Odyssey was used to analyze levels over a linear dynamic range.

## Constructs for protein expression in *E. coli*

The plasmids for 6His-PreScission-DCX-EGFP-StrepII (Addgene #83918), kif5b(1-560)-EGFP-6His (Addgene #15219), and Sfp-6His (Addgene #75015) were ordered from Addgene. The plasmid for JIP3 was a gift from Cavalli lab (Valeria Cavalli, Department of Anatomy and Neurobiology, Washington University in St Louis, School of Medicine, St Louis, MO) (*Sun et al., 2011*). For DCX, EGFP was replaced by a ybbR-tag using Q5 mutagenesis. For kif5b, the sequence encoding amino acids 1–490 was amplified with NdeI and EcoRI overhangs and inserted into a modified backbone based on pSNAP-tag(T7)2 (NEB #N9181S) before a SNAPf-EGFP-6His tag (*Budaitis et al., 2021*). For JIP3, the sequence encoding amino acids 3–240 (or 3–548) was amplified with NdeI-6His and EcoRI overhangs.

The first two amino acids are Met in JIP3, which were therefore skipped because a 6His-tag was inserted at the N-terminus. The amplified sequence was then inserted into a modified backbone based on pSNAP-tag(T7)2 before a HaloTag-StrepII tag. All constructs were verified by restriction enzyme digestion and DNA sequencing.

## Protein expression in *E. coli*

Protein expression in *E. coli* was done as previously described (*Budaitis et al., 2021*). Briefly, a plasmid was transformed into BL21-CodonPlus(DE3)-RIPL-competent cells (Agilent #230280), and a single colony was inoculated in 1 ml of TB with 50 µg/ml of chloramphenicol and 25 µg/ml of carbenicillin or 15 µg/ml of kanamycin in the case of Sfp. The culture was shaken at 37°C overnight, and then inoculated into 400 ml of TB, which was shaken at 37°C for 5 hr, and subsequently cooled down to 18°C. IPTG was added to the culture to a final concentration of 0.1 mM, and the expression was induced overnight at 18°C with shaking. The culture was harvested by centrifugation at 3000 rcf for 10 min. Following the removal of the supernatant, the cell pellet was resuspended in 5 ml of B-PER complete (Thermo Scientific #89821) supplemented with 4 mM MgCl$_2$, 2 mM EGTA, 0.2 mM ATP, 2 mM DTT, and 2 mM PMSF. The cell suspension was then flash-frozen and stored at –80°C.

## Purification and labeling of Sfp, JIP3, DCX, and KIF5B

The purification of *E. coli* expressed protein was done as previously described (*Budaitis et al., 2021*). For the JIP3 and DCX constructs, a two-step purification was performed. For Sfp and kif5b, only the His-tag purification was performed. Briefly, the cell pellet was thawed at 37°C, and then nutated at room temperature for 20 min. The lysate was dounced for 10 strokes on ice and cleared via centrifugation at 80,000 × rpm (260,000 × *g*) for 10 min at 4°C using a TLA 110 rotor (Beckman) in a tabletop Beckman ultracentrifuge. At the same time, 500 µl of the Ni-NTA slurry (Roche cOmplete His-Tag purification resin) was washed with 2 × 1 ml of wash buffer (WB, 50 mM HEPES, 300 mM KCl, 2 mM MgCl$_2$, 1 mM EGTA, 1 mM DTT, 0.1 mM ATP, 1 mM PMSF, 0.1% [w/v] Pluronic F-127, pH 7.2) in a 10 ml column (Bio-Rad #7311550). After the centrifugation, the supernatant was loaded into the column and allowed to flow through the resin by gravity. The resin was then washed with 3 × 2 ml of WB.

For Halo-tag labeling, halo-tag ligand was added to final 10 µM, and the resin was incubated at room temperature for 10 min. For ybbR-tag labeling, the CoA-dye ligand for ybbR-tag was generated by reacting coenzyme A (CoA) with a dye containing a maleimide group in 1:1 ratio at room temperature for 30 min. The final product was quenched with 50 mM DTT, aliquoted, flash-frozen, and stored at –80°C. To label the ybbR-tag, CoA-dye and Sfp was added to the resin to a final concentration of 10 µM. The resin was nutated at 4°C for 3 hr. After the labeling, the resin was washed with 3 × 3 ml of WB and eluted with Ni-NTA elution buffer (Ni-EB, 50 mM HEPES, 150 mM KCl, 2 mM MgCl$_2$, 1 mM EGTA, 1 mM DTT, 0.1 mM ATP, 1 mM PMSF, 0.1% [w/v] Pluronic F-127, 250 mM imidazole, pH 7.2). For Sfp and kif5b, the protein was aliquoted, flash-frozen, and stored at –80°C until further usage. For JIP3 and DCX, the concentrated fraction was pooled and flown through 1 ml of streptactin slurry (IBA #2-1201) that had been washed with 2 × 1 ml WB. The resin was then washed with 3 × 2 ml WB, and then eluted with streptactin elution buffer (St-EB, 50 mM HEPES, 150 mM KCl, 2 mM MgCl$_2$, 1 mM EGTA, 1 mM DTT, 0.1 mM ATP, 1 mM PMSF, 0.1% [w/v] Pluronic F-127, 2.5 mM dethiobiotin, pH 7.2). The concentrated fraction was pooled and further concentrated via centrifugation using Amicon 0.5 ml 10 kDa unit. The protein was verified on a PAGE gel, and the concentration was determined using Bradford assay.

## Constructs for protein expression in insect cells

The pFastBac plasmid with codon-optimized full-length human dynein was a gift from the Carter lab (MRC Laboratory of Molecular Biology, Francis Crick Avenue, Cambridge, UK) (*Schlager et al., 2014*). The pFastBac plasmid that encodes tail-truncated human dynein (amino acids 1320–4646 of DYNC1H1) was a gift from the Reck-Peterson Lab (Department of Cellular and Molecular Medicine, University of California, San Diego, CA) (*Htet et al., 2020*).

## Protein expression in insect cells

Full-length human dynein and tail-truncated human dynein were expressed in Sf9 cells as described previously (*Schlager et al., 2014*; *Htet et al., 2020*). Briefly, the pFastBac plasmid containing

full-length human dynein or tail-truncated dynein was transformed into DH10Bac-competent cells (Gibco, #10361012) with heat shock at 42°C for 45 s followed by incubation at 37°C for 4 hr in S.O.C. medium (Gibco, #15544034). The cells were then plated onto LB agar plates containing kanamycin (50 µg/ml), gentamicin (7 µg/ml), tetracycline (10 µg/mL), BluoGal (100 µg/mL), and isopropyl-β-D-thiogalactoside (IPTG; 40 µg/mL), and positive clones were identified by a blue/white color screen after 36 hr. Bacmid DNA was extracted from overnight culture using an isopropanol precipitation method with QIAGEN buffer (QIAGEN, #27104) as described previously (*Schlager et al., 2014*). To generate baculovirus for Sf9 insect cell transfection, 2 ml of Sf9 cells at $0.5 \times 10^6$ cells per ml in six-well plates (Corning, #3516) were transfected with 2 µg of fresh bacmid DNA and 6 µl of FuGene HD transfection reagent (Promega, E2311) according to the manufacturer's instruction. The cells were incubated for 4 days, and the supernatant containing V0 virus was collected. To generate V1 virus, 0.5 ml of V0 virus was used to transfect 50 ml of Sf9 cells at $1.5 \times 10^6$ cells per ml. The supernatant containing V1 virus was collected by centrifugation at $200 \times g$ for 5 min at 4°C after 3 days. The V1 virus was stored at 4°C in the dark until use. For protein expression, 5 ml of the V1 virus was used to transfect 500 ml Sf9 cells at $2 \times 10^6$ cells per ml. After 60 hr incubation, cells were collected by centrifugation at $3000 \times g$ for 10 min at 4°C. The cell pellet was resuspended in 15 ml ice-cold PBS and centrifuged again. The supernatant was then removed, and the cell pellet was flash-frozen in liquid nitrogen and stored at –80°C.

## Purification and labeling of tail-truncated and full-length human dynein

Full-length dynein and tail-truncated dynein were purified from frozen Sf9 pellets as described previously (*Schlager et al., 2014*; *Htet et al., 2020*). Frozen pellets from 500 ml insect cell culture were thawed on ice and resuspended in lysis buffer (50 mM HEPES pH 7.4, 100 mM NaCl, 1 mM DTT, 0.1 mM ATP, 10% [v/v] glycerol, 2 mM PMSF) supplemented with one protease inhibitor cocktail tablet (cOmplete-EDTA free, Roche, #11836170001) to a final volume of 50 ml. Cells were then lysed using a Dounce homogenizer with 20 strokes. The lysate was cleared by centrifugation at $279,288 \times g$ for 10 min at 4°C using a Beckman Coulter tabletop centrifuge unit. The clarified supernatant was incubated with 3 ml of IgG Sepharose 6 Fast Flow beads (Cytiva, #17096901) for 4 hr with rotation. After incubation, the protein-bound IgG beads were transferred to a gravity flow column and washed with 100 ml lysis buffer and 100 ml TEV buffer (50 mM Tris–HCl pH 8.0, 250 mM potassium acetate, 2 mM magnesium acetate, 1 mM EGTA, 1 mM DTT, 0.1 mM Mg-ATP and 10% [v/v] glycerol). To fluorescently label the carboxy-terminal SNAPf tag of full-length human dynein, dynein-coated beads were incubated with 5 µM SNAP-Cell TMR (New England Biolabs, #S9105S) in the column for 10 min at room temperature. The beads were then washed with 100 ml TEV buffer at 4°C to remove unbound dyes. Subsequently, the beads were resuspended in TEV buffer (final volume 5 ml) with 100 µl TEV protease (New England Biolabs, #P8112S) and incubated at 4°C on a roller overnight. After TEV cleavage, the beads were removed and protein of interest was concentrated using a 100 kDa molecular weight cut-off (MWCO) concentrator (Millipore, #Z648043) to 1 ml and flash-frozen in liquid nitrogen.

## Microtubule polymerization

Microtubule polymerization was performed as described before (*Rao et al., 2018*). Briefly, 2 µl of 10 mg/ml unlabeled tubulin (Cytoskeleton) was mixed with 2 µl of 1 mg/ml biotin-tubulin and 2 µl of 1 mg/ml dye-labeled tubulin on ice. 0.5 µl of 10 mM GTP was added to the mixture, and the mixture was incubated at 37°C for 20 min. Afterward 0.7 µl of 0.2 mM taxol (in DMSO) was added, and the solution was incubated at 37°C for another 15 min. The unincorporated tubulin was removed by centrifuging through a glycerol cushion (80 mM PIPES, 2 mM $MgCl_2$, 1 mM EGTA, 60% glycerol, 10 µM taxol, 1 mM DTT, pH 6.8) at $80,000 \times rpm$ ($250,000 \times g$) for 5 min at room temperature using TLA 100 motor (Beckman) in a tabletop Beckman ultracentrifuge. The supernatant was discarded, and the pellet was resuspended in 12 µl resuspension buffer (80 mM PIPES, 2 mM $MgCl_2$, 1 mM EGTA, 10% glycerol, 10 µM taxol, 1 mM DTT, pH 6.8) to obtain a final 2 mg/ml MT concentration for the TIRF assay. For the MT-binding and -release assay, 5 µl of 10 mg/ml unlabeled tubulin was used to polymerize the MTs, and the pellet was resuspended in 10 µl of the resuspension buffer to obtain a final 5 mg/ml MT concentration.

## Microtubule-binding and -release assay of kif5b and single-head human dynein

Impaired/inactive motors were removed by an MT-binding and -release assay as described before (*Budaitis et al., 2021*; *Rao et al., 2019*). Briefly, 50 µl of protein solution was exchanged into a binding buffer (30 mM HEPES, 50 mM KCl, 2 mM MgCl$_2$, 1 mM EGTA, 10% glycerol, 1 mM DTT, 0.1 mM AMPPNP, pH 7.2) using Zeba 7 kDa unit (Thermo Scientific #89882). The protein solution was warmed to room temperature, and taxol was added to final 20 µM concentration. For kif5b, AMPPNP was also added to a final 1 mM concentration. 3 µl of the 5 mg/ml MT stock was added to the protein solution and then mixed well. The solution was then carefully layered on top of 100 µl of glycerol cushion for kif5b (80 mM PIPES, 2 mM MgCl$_2$, 1 mM EGTA, 60% glycerol, 10 µM taxol, 1 mM DTT, pH 6.8) or sucrose cushion for dynein (30 mM HEPES, 50 mM KCl, 2 mM MgCl$_2$, 10% glycerol, 25% w/v sucrose, 10 µM taxol, 1 mM DTT, pH 7.2) in a TLA 100 rotor (Beckman), and centrifuged at 45,000 × rpm (80,000 × *g*) for kif5b or 80,000 × rpm (250,000 × *g*) for dynein at room temperature for 10 min. Afterward the supernatant was removed and the pellet was washed with 2 × 20 µl wash buffer (30 mM HEPES, 50 mM KCl, 2 mM MgCl$_2$, 10% glycerol, 10 µM taxol, 1 mM DTT, pH 7.2). The pellet was then resuspended in 47 µl of high-salt release buffer (HSRB, 30 mM HEPES, 300 mM KCl, 2 mM MgCl$_2$, 10% glycerol, 10µM taxol, 1 mM DTT, pH 7.2), and 3 µl of 100 mM ATP was added to the solution. The solution was centrifuged at 40,000 × rpm (60,000 × *g*) for 5 min, and the supernatant was aliquoted, flash-frozen, and stored at –80°C for further usage.

## DDJ complex assembly

DDJ complex was assembled following a published protocol (*Urnavicius et al., 2018*). Briefly, dynein, dynactin (a gift from the laboratory of Andrew Carter, MRC), and JIP3 were mixed on ice in 1:1:1 ratio (final concentration of 200 nM each) and incubated for 1 hr on ice in the dark. For DDJ complex formation in the presence of N-DCX, N-DCX was added in equal amount as dynein.

## TIRF motility assay

The TIRF motility assay was performed as described before (*Budaitis et al., 2021*). Briefly, a coverslip was cleaned using ethanol and assembled into a flow chamber. 10 µl of 0.5 mg/ml biotin-BSA was introduced into the flow chamber, and the flow chamber was incubated at room temperature for 10 min in a humidity chamber. The chamber was then washed with 3 × 20 µl of blocking buffer (BB, 80 mM PIPES, 2 mM MgCl$_2$, 1 mM EGTA, 10 µM taxol, 1% [w/v] Pluronic F-127, 2 mg/ml BSA, 1 mg/ml α-casein, pH 6.8), and incubated for 10 min. The solution in the chamber was completely removed using vacuum, and 10 µl of 0.25 mg/ml streptavidin was flown in and incubated at room temperature for 10 min. The chamber was then washed with 3 × 20 µl of BB, and the solution was completely removed afterward. 0.5 µl of 0.2 mg/ml fluorescently labeled MTs was diluted in 19.5 µl of BB and flown into the chamber. The chamber was then washed with 2 × 20 µl BB and 20 µl of motility buffer (MB, 60 mM HEPES, 50 mM KAc, 2 mM MgCl$_2$, 1 mM EGTA, 0.5% [w/v] Pluronic F-127, 10 µM taxol, 1 mM DTT, 5 mg/ml BSA, 1 mg/ml α-casein, pH 7.2). 1 µl of 100 mM ATP, 1 µl of 50 mM biotin, and 1 µl of oxygen scavenger system were added to 46 µl of MB, and 1 µl of 10 nM DDJ complex was added subsequently. For DCX and N-DCX experiments, DCX was added to a final 10 nM in the final solution. The solution was mixed well and flown into the chamber. The chamber was then sealed with vacuum grease. The acquisition time was 200 ms per frame, and a total of 600 frames was taken for each movie. The data was analyzed using a custom-built MATLAB software. Alternatively, ImageJ could be used to generate and analyze the kymographs, which yields the same results. The statistical analysis and data visualization were performed using Prism.

## TIRF gliding assay

A slide chamber was assembled as described above. 10 µl of 0.1 mg/mL anti-GFP antibody (YenZym) was introduced into the chamber, which was then incubated in a humidity chamber for 10 min. The chamber was washed with 3 × 20 µl BB and 20 µl of MB. 1 µl of MTBR fraction of single-head human dynein was diluted in 19 µl of MB, and the solution was flown into the chamber and incubated for 2 min. The chamber was washed with 3 × 20 µl of MB to remove unbound dynein. 1 µl of 100 mM ATP, 0.5 µl of 0.2 mg/ml MTs, and 1 µl of oxygen scavenger system were added to 47.5 µl of MB, which

was flown into the chamber. The chamber was sealed with vacuum grease. The imaging condition and analysis were done as described above.

## TIRF landing assay

A slide chamber was constructed and MTs were immobilized on the coverslip as in the TIRF mobility assay. Single-head human dynein was diluted to appropriate concentration in the motility buffer. 2 mM ATP was used for the ATP state; 2 mM ADP and 0.05 U/µl hexokinase were used for the ADP state; 2 mM AMPPNP was used for the AMPPNP state. The image acquisition and data analysis were done as in the TIRF motility assay.

## Statistical analysis

Statistical analyses were performed using GraphPad Prism 7.0 software (GraphPad Software Inc, San Diego, CA). All data are presented as the mean ± SEM of at least three independent experiments. Statistical significance was determined using one-way analysis of variance (ANOVA) followed by Tukey's test if more than two groups were analyzed. Two-tailed test and Student's $t$-test were used to compare two groups. $p < 0.05$ was considered significant (*$p < 0.05$; #$p < 0.01$, if not specified otherwise).

## Acknowledgements

The authors thank Lisa Baker and Julian Curiel for help with the editing of the manuscript. LR and AG thank Andrew Carter (LMB Cambridge) for generously providing purified porcine brain dynactin. This work was supported by the Brain and Behavior Research Foundation (JSL and MT), the Whitehall Foundation (JSL), the National Natural Science Foundation of China (81971425 and 81871035) (XF and PL), the Zhejiang Provincial Natural Science Foundation of China (LZ09H090001 and LY20H040002) (PL and XF), the National Institute of Health (NIH) grants R01GM098469 and R01NS114636 (LR, XL, and AG), and the NIH grant RO1NS104428-01 (JSL).

## Additional information

### Funding

| Funder | Grant reference number | Author |
|---|---|---|
| National Natural Science Foundation of China | 81971425 | Xiaoqin Fu |
| National Natural Science Foundation of China | 81871035 | Peijun Li |
| Natural Science Foundation of Zhejiang Province | LZ09H090001 | Peijun Li |
| Natural Science Foundation of Zhejiang Province | LY20H040002 | Xiaoqin Fu |
| National Institutes of Health | R01GM098469 | Arne Gennerich |
| National Institutes of Health | R01NS114636 | Arne Gennerich |
| National Institutes of Health | RO1NS104428-01 | Judy Shih-Hwa Liu |
| Brain and Behavior Research Foundation | | Judy Shih-Hwa Liu |
| Whitehall Foundation | | Judy Shih-Hwa Liu |

| Funder | Grant reference number | Author |
|--------|------------------------|--------|

The funders had no role in study design, data collection and interpretation, or the decision to submit the work for publication.

## Author contributions

Xiaoqin Fu, Arne Gennerich, Judy Shih-Hwa Liu, Conceptualization, Resources, Data curation, Software, Formal analysis, Supervision, Funding acquisition, Validation, Investigation, Visualization, Methodology, Writing – original draft, Project administration, Writing – review and editing; Lu Rao, Conceptualization, Data curation, Formal analysis, Validation, Investigation, Visualization, Methodology, Writing – original draft, Writing – review and editing; Peijun Li, Conceptualization, Data curation, Formal analysis, Funding acquisition, Validation, Visualization, Writing – review and editing; Xinglei Liu, Validation, Investigation, Methodology; Qi Wang, Validation, Methodology; Alexander I Son, Validation, Visualization, Methodology

## Author ORCIDs

Xiaoqin Fu http://orcid.org/0000-0001-6354-8960
Arne Gennerich http://orcid.org/0000-0002-8346-5473

## Ethics

All animal procedures were approved by the Committee on the Ethics of Animal Experiments of Wenzhou Medical University (Permit number: wydw2019-0723).

## Decision letter and Author response

Decision letter https://doi.org/10.7554/eLife.82218.sa1
Author response https://doi.org/10.7554/eLife.82218.sa2

## Additional files

### Supplementary files

- MDAR checklist

### Data availability

All data generated or analysed during this study are included in the manuscript and supporting file; Source Data files have been provided for figure 2, 3, 4, figure 2-figure supplement 1 and table 1.

The following dataset was generated:

| Author(s) | Year | Dataset title | Dataset URL | Database and Identifier |
|-----------|------|---------------|-------------|--------------------------|
| Fu X, Rao L, Li P, Liu X, Wang Q, Son A, Gennerich A, Shih-Hwa Liu J | 2022 | Doublecortin and JIP3 are neural-specific counteracting regulators of dynein-mediated retrograde trafficking | https://dx.doi.org/10.5061/dryad.vmcvdncwt | Dryad Digital Repository, 10.5061/dryad.vmcvdncwt |

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
