## [Editor Report]

In their article, Fu, Rao et al. explore the mechanisms by which the microtubule-associated protein, doublecortin (DCX), functions in regulating retrograde transport in neurons. They reconstitute a dynein-dynactin-JIP3 complex, validating JIP3 as a bona fide adaptor, and find that DCX disrupts the initial on-rate and therefore transport of this processive complex both in vivo and in vitro. This mechanism will be valuable in understanding how mutations in DCX cause lissencephaly, and this solid article will be of interest to those in the cytoskeletal and neurobiology fields.

---

## [Decision Letter]

**Decision letter after peer review:**

Thank you for submitting your article "Doublecortin and JIP3 are neural-specific counteracting regulators of dynein-mediated retrograde trafficking" for consideration by *eLife*. Your article has been reviewed by 2 peer reviewers, and the evaluation has been overseen by a Reviewing Editor and Anna Akhmanova as the Senior Editor. The following individual involved in the review of your submission has agreed to reveal their identity: Paulomi Sanghavi (Reviewer #1).

Essential revisions:

1) Both Reviewers indicated that showing the effects of the knockdown conditions on anterograde transport by assaying a different cargo would strengthen the paper. Please refer to the comments for more details.

2) As Reviewer 2 points out, it would add mechanistic insight to determine if DCX reduces the MT binding affinity of dynein, or if it increases dynein detachment. This can probably be extracted from your current in vitro dataset.

3) Although the authors do not need to show that dynein and DCX interact directly to warrant publication if you do not show this, please add a sentence within the manuscript about how you have not assayed this interaction directly and therefore do not know if this is an indirect or direct interaction.

4) For this paper, it is important to distinguish the dynein-binding and microtubule-binding functions of DCX in affecting retrograde transport. As Reviewer 1 suggests, can the authors determine if the DCX dynein-binding mutant is associated with microtubules? Does this mutant rescue any phenotypes in a knockout background?

5) Please address both reviewers' comments about adding more discussion or explanation of certain results. These textual changes will help the reader in understanding the results, and the overall model, and place them within the context of the field.

*Reviewer #1 (Recommendations for the authors):*

Following are the areas which I think might require more clarification for a better understanding of the work.

1. In Figure 3C we see less dynein in the supernatant even when there is hardly an increase in dynein levels in the pellet. How do we explain this?

2. Similarly in Figure 3D, we see more dynein binding to microtubule pellet and the authors conclude that the presence of DCX prevents dynein interaction with microtubules. Does the presence of DCX reduce the binding affinity of dynein to MTs or does it increase dynein detachment? It might help to make this distinction to better understand how DCX functions.

3. Do DCX knockouts show increased retrograde transport of TrkB vesicles? Do we see reduced anterograde transport mediated by kinesin-3 of these vesicles/other cargoes in this background?

4. C-terminus of DCX is shown to increase DCX association with microtubules but reduces its binding to dynein. How does the protein function? Does the C-terminus fold over when it needs to bind microtubules versus dynein? And attain a different conformation when microtubule binding is required? Speculations?

5. DCX microtubule binding mutants can bind dynein but not rescue the knockout phenotype so that means dynein binding function is dependent on its microtubule binding. How can we explain that?

6. Does DCX interact with Dynein directly?

7. Does DCX dynein binding mutant associate with microtubules? How much rescue of phenotype do we obtain upon adding this mutant in a dcx/dlck1 knockout background? This might help us to understand the contribution of each of these two functions- dynein binding and microtubule binding of DCX in affecting retrograde transport.

*Reviewer #2 (Recommendations for the authors):*

(i) I request authors to update the publication list. The study from Reto Gassmann's lab is referenced as being on biorxiv but is now published in JCB.

(ii) Do authors have an explanation of effects only on run lengths but not on total flux of retrograde TrkB cargo in DCX-/- neurons? In general, in all knockdown, it would have been useful to show the flux and other transport data in both anterograde and retrograde directions. This will strengthen the conclusions that DCX effects on the cargos assayed are specifically associated only with retrograde movement.

(iii) JIP3 is an adaptor for Kinesin-1 and Dynein. It would be nice to include a short discussion of what if any role DCX might play with JIP3 on cargos that recruit both Dynein and Kinesin-1.

---

## [Author Response]

Essential revisions:1) Both Reviewers indicated that showing the effects of the knockdown conditions on anterograde transport by assaying a different cargo would strengthen the paper. Please refer to the comments for more details.

According to our results, about 10% of the imaged TrkB vesicles were transported in anterograde direction. We have analyzed those anterogradely transported TrkB vesicles to determine whether DCX has effects on anterograde TrkB transport. After analyzing anterogradely transported TrkB vesicles from neurons recorded from 4 independent experiments in each group, we did not see significant differences in both velocity and run length of anterograde transported TrkB vesicles between wild type (39 vesicles from 27 neurons, n=4) and *Dcx-/y* neurons (43 vesicles from 27 neurons, n=4). The results are shown in Figure 1—figure supplement 1C. Meanwhile, as we studied the anterograde transport of the KIF1A-driven Vamp2 cargo in DCX knockout neurons before (Liu et al. 2012, Molecular Cell), we refrained from studying another anterogradely transported cargo and discuss our anterograde transport results in the revised manuscript.

2) As Reviewer 2 points out, it would add mechanistic insight to determine if DCX reduces the MT binding affinity of dynein, or if it increases dynein detachment. This can probably be extracted from your current in vitro dataset.

We thank the reviewer for this suggestion. We agree that the effects of DCX on the MT-binding affinity of dynein would add mechanistic insight into the regulation of dynein transport by DCX. We therefore performed a total internal reflection fluorescence (TIRF) assay to determine the relative microtubule on- and off-rates of dynein in the absence or presence of DCX. The results show that while DCX does not affect the dwell time (off-rate) of single-headed dynein construct, it reduces the landing rate (on-rate) of dynein in both the ADP and AMP-PnP states (new Figure 5—figure supplement 5). This result implies that DCX interferes with the initial (weak) association of dynein with MTs but does not displace dynein from the MTs once dynein is bound. That the effect of DCX on dynein’s MT landing rate is rather small agrees with the significant but minor effect of DCX on the velocity of MT gliding by dynein. This also could explain why there is no obvious effect of DCX on the movement of DDB complexes at the single-molecule level (Monroy et al., 2020).

3) Although the authors do not need to show that dynein and DCX interact directly to warrant publication if you do not show this, please add a sentence within the manuscript about how you have not assayed this interaction directly and therefore do not know if this is an indirect or direct interaction.

Previous studies show that DCX associates with the dynein motor complex (Tanaka et al., 2004; Taylor et al., 2000). To further probe the interaction of DCX with dynein, we used recombinant HA (negative control) and HA-DCX constructs as bait to pull-down DCX-interacting proteins from mouse brains using mass-spectrometry analysis. Table 1 shows that our analysis identified cytoplasmic dynein 1 intermediate chain 2 and dynactin subunit 5 as DCX-interacting proteins. Although this analysis does not rule out the possibility that other unknown proteins are required for the interaction of DCX with IC2 and the dynactin subunit 5, it is very likely that dynein and DCX interact directly as DCX affects the composition of DDJ motor complexes that are formed from recombinantly expressed proteins (Figure 5C). We added this paragraph in the Result (line 150-159).

4) For this paper, it is important to distinguish the dynein-binding and microtubule-binding functions of DCX in affecting retrograde transport. As Reviewer 1 suggests, can the authors determine if the DCX dynein-binding mutant is associated with microtubules? Does this mutant rescue any phenotypes in a knockout background?

Please see our detailed response to this comment under Q7 of reviewer 1.

5) Please address both reviewers' comments about adding more discussion or explanation of certain results. These textual changes will help the reader in understanding the results, and the overall model, and place them within the context of the field.

We revised the discussion as requested.

Reviewer #1 (Recommendations for the authors):Following are the areas which I think might require more clarification for a better understanding of the work.1. In Figure 3C we see less dynein in the supernatant even when there is hardly an increase in dynein levels in the pellet. How do we explain this?

The important point of this experiment is to measure the proportion of DIC in the pellet (P). To do this, we calculated the ratios of P/(P+S) for different groups. Although the difference is not big, the proportion of DIC in P from DCX knockout is still significantly higher (P=0.046) than that from WT (chart of Figure 3C). This difference is even bigger between DCX/DCLK1 double knockout and WT as shown in Figure 3D (P=0.016). Since DCLK1 is a functionally redundant and structurally related paralog of DCX in mice, it is not surprising that the absence of both DCX and DCLK1 have further effects on DIC association with MT. To address this comment further, we have replaced Figure 3C with a more representative picture.

2. Similarly in Figure 3D, we see more dynein binding to microtubule pellet and the authors conclude that the presence of DCX prevents dynein interaction with microtubules. Does the presence of DCX reduce the binding affinity of dynein to MTs or does it increase dynein detachment? It might help to make this distinction to better understand how DCX functions.

We thank the reviewer for this suggestion. We agree that the effects of DCX on the MT-binding affinity of dynein would add mechanistic insight into the regulation of dynein transport by DCX. We performed a TIRF assay to determine the relative MT on- and off-rates of dynein in the absence or presence of DCX. Our results show that while DCX does not affect dynein’s MT dwell time (off-rate), it reduces the landing rate (on-rate) of dynein in both the ADP and AMP-PnP states (Figure 5—figure supplement 5). This result implies that DCX interferes with the initial (weak) association of dynein with MTs but does not displace dynein from the MTs once dynein is bound. That the effect of DCX on dynein’s MT landing rate is rather small agrees with the significant but minor effect of DCX on the velocity of MT gliding by dynein. This also could explain why there is no obvious effect of DCX on the movement of DDB complexes at the single-molecule level (Monroy et al., 2020).

3. Do DCX knockouts show increased retrograde transport of TrkB vesicles? Do we see reduced anterograde transport mediated by kinesin-3 of these vesicles/other cargoes in this background?

Yes, Figure 1D and 1E show increased retrograde transport of TrkB vesicles in *Dcx* knockout neurons compared to the controls, and we showed previously that the anterograde transport of Vamp2 driven by kinesin-3 (KIF1A) is significantly decreased in DCX knockout neurons (Liu et al., 2012). In contrast to these observations, we find that the ~10% of anterogradely moving TrkB vesicles do not show a significant difference in velocity and run length between wild type (39 vesicles from 27 neurons, n=4) and *Dcx-/y* neurons (43 vesicles from 27 neurons, n=4). As both kinesin-1(KIF5) and KIF1A drive the anterograde transport of TrkB (Arimura et al., 2009; Huang et al., 2011; Sun et al., 2017; Zahavi et al., 2021), this suggests that DCX’s promoting effect on KIF1A might counteract DCX’s inhibitory effect on KIF5 (Figure 5C) so that the combined effects are cancelling each other out, so that no differences in anterograde transport of TrkB occur with DCX deficiency. We provide our new results and this discussion in our revised manuscript (Figure 1—figure supplement 1C and Discussion from line 441-line 474).

4. C-terminus of DCX is shown to increase DCX association with microtubules but reduces its binding to dynein. How does the protein function? Does the C-terminus fold over when it needs to bind microtubules versus dynein? And attain a different conformation when microtubule binding is required? Speculations?

Yes, the C-terminus of DCX increases DCX association with MTs based on previous reports and our new results. DCX forms dimerized or higher-order structures on MTs and this self-association is MT-dependent (Moores et al., 2006; Rafiei et al., 2022), while in solution, DCX rarely oligomerizes by itself (Moores et al., 2006). Therefore, the conformation of DCX changes when DCX associates with MTs and the binding to MTs likely relieves DCX’s “closed” conformation and thereby exposes/frees DCX’s C-terminus. On the other hand, we show that the C-terminal domain decreases DCX-dynein association while the C-terminal domain itself does not bind to dynein or MTs. In contrast to the C-terminal domain, the N-terminal DCX domain associates with both MTs and dynein. This suggests that unlike MTs, dynein does not alter the conformation of DCX following binding. We hypothesize that the association of DCX with MTs and dynein is a dynamic phenomenon. When the association of DCX with MTs increases, DCX association with dynein decreases, and vice versa. This dynamic process is likely regulated through conformational changes of C-terminus (e.g., through interactions of the C-terminal domain with the N-terminal domain, an interaction that could be regulated by kinases through phosphorylation of residues in the C-terminal domain). We discuss all these possibilities in our revised manuscript.

5. DCX microtubule binding mutants can bind dynein but not rescue the knockout phenotype so that means dynein binding function is dependent on its microtubule binding. How can we explain that?

The association of DCX with MTs and dynein is likely highly dynamic and competitive as we discussed above. The cooperative binding of DCX to MTs in vivo increases the local DCX concentration, which in turn increases the chances that a DCX molecule associates with a nearby DDJ complex, causing the second dynein to dissociate and thereby reducing the velocity of the DDJ complex. As the A71S and T203R mutants cannot accumulate locally on MTs, they may have a reduced chances of binding to the DDJ complex and thus show a significantly reduced inhibitory effect on retrograde transport. This explanation is incorporated in the revised manuscript from line 486 to line 500.

6. Does DCX interact with Dynein directly?

Previous studies show that DCX associates with the dynein motor complex (Tanaka et al., 2004; Taylor et al., 2000). To further probe the interaction of DCX with dynein, we used recombinant HA (negative control) and HA-DCX constructs as bait to pull-down DCX-interacting proteins from mouse brains using mass-spectrometry analysis. Table 1 shows that our analysis identified cytoplasmic dynein 1 intermediate chain 2 and dynactin subunit 5 as DCX-interacting proteins. Although this analysis does not rule out the possibility that other unknown proteins are required for the interaction of DCX with IC2 and the dynactin subunit 5, it is very likely that dynein and DCX interact directly as DCX affects the composition of DDJ motor complexes that are formed from recombinantly expressed proteins (Figure 5C). We added this paragraph in the Result (line 150-158).

7. Does DCX dynein binding mutant associate with microtubules? How much rescue of phenotype do we obtain upon adding this mutant in a dcx/dlck1 knockout background? This might help us to understand the contribution of each of these two functions- dynein binding and microtubule binding of DCX in affecting retrograde transport.

1) Yes, our data show that DCX dynein binding mutant (N-DCX), which has stronger binding affinity with dynein, still associates with MTs but with weaker affinity. As we explained to Q4, the C-terminal domain of DCX does not bind to dynein or MTs, it is the N-terminal DCX domain that associates with both MTs and dynein. Compared to full length DCX, the N-DCX mutant (without the C-terminal domain) has a stronger affinity for dynein (Figure 2—figure supplement 1A-C), while it has a weaker affinity for MTs. This is consistent with our hypothesis that when the association of DCX with MTs increases, DCX association with dynein decreases, and vice versa.

2) Introducing N-DCX either into DCX knockout neurons (Figure 2C-D and Figure 2—figure supplement 2A-B) or WT neurons (Figure 2F-G and Figure 2—figure supplement 2C-D) decreases the retrograde transport of TrkB to a greater extent than FL-DCX. Therefore, this DCX dynein binding mutant (N-DCX) has stronger rescue effect on the TrkB retrograde transport phenotype in *dcx/dlck1* knockout background than wild type DCX.

3) DCX mutants (A71S and T203R) lose their ability to associate with MTs, but these mutants can still bind dynein, whether with higher affinity is not known. The reason that they cannot rescue the knockout phenotype is presented under Q5. That means DCX-MT binding is necessary for DCX’s effects on dynein-mediated retrograde transport.

4) We currently do not know other DCX mutants which decrease the interaction with dynein complex. It would be a future direction to further confirm our current conclusion and unravel the in-depth mechanism around DCX/dynein/MT.

Reviewer #2 (Recommendations for the authors):(i) I request authors to update the publication list. The study from Reto Gassmann's lab is referenced as being on biorxiv but is now published in JCB.

We have updated the reference in the revised manuscript.

(ii) Do authors have an explanation of effects only on run lengths but not on total flux of retrograde TrkB cargo in DCX-/- neurons? In general, in all knockdown, it would have been useful to show the flux and other transport data in both anterograde and retrograde directions. This will strengthen the conclusions that DCX effects on the cargos assayed are specifically associated only with retrograde movement.

We thank the reviewer for this suggestion. We have now analyzed the anterogradely transported TrkB vesicles as well. The results of these analyses are shown in Figure 1—figure supplement 1C. We did not see significant differences in both velocity and run length of anterogradely transported TrkB vesicles between wild type (39 vesicles from 27 neurons, n=4) and *Dcx-/y* neurons (43 vesicles from 27 neurons, n=4). We have also added discussion about this data to the revised manuscript (from line 435 to line 474). In addition, we are also providing the percentages of TrkB vesicles transported in retrograde direction and anterograde direction, as well as the percentages of stationary TrkB vesicles in different neurons (see Figure 1—figure supplement 1B, Figure 2—figure supplement 2A and 2C, and Figure 3—figure supplement 1A). Finally, we are providing the distribution of run lengths and velocities of retrogradely transported TrkB vesicles in different neurons (see Figure 1—figure supplement 1D, Figure 2—figure supplement 2B and 2D, Figure 3—figure supplement 1B, and Figure 4—figure supplement 1).

(iii) JIP3 is an adaptor for Kinesin-1 and Dynein. It would be nice to include a short discussion of what if any role DCX might play with JIP3 on cargos that recruit both Dynein and Kinesin-1.

We thank the reviewer for this suggestion. TrkB is a cargo recruited by both dynein and kinesin-1 (KIF5), and JIP3 is indeed involved in the recruitment of both dynein and KIF5. We have analyzed the anterograde transport of TrkB in wild type neurons and DCX deficient neurons, and are presenting the new data in Figure 1—figure supplement 1C and have added discussion (from line 455 to line 462) to explain DCX’ effects on anterograde transport of TrkB to the revised manuscript.

References:

Arimura, N., T. Kimura, S. Nakamuta, S. Taya, Y. Funahashi, A. Hattori, A. Shimada, C. Menager, S. Kawabata, K. Fujii, A. Iwamatsu, R.A. Segal, M. Fukuda, and K. Kaibuchi. 2009. Anterograde transport of TrkB in axons is mediated by direct interaction with Slp1 and Rab27. Dev Cell. 16:675-686.

Huang, S.H., S. Duan, T. Sun, J. Wang, L. Zhao, Z. Geng, J. Yan, H.J. Sun, and Z.Y. Chen. 2011. JIP3 mediates TrkB axonal anterograde transport and enhances BDNF signaling by directly bridging TrkB with kinesin-1. J Neurosci. 31:10602-10614.

Liu, J.S., C.R. Schubert, X. Fu, F.J. Fourniol, J.K. Jaiswal, A. Houdusse, C.M. Stultz, C.A. Moores, and C.A. Walsh. 2012. Molecular basis for specific regulation of neuronal kinesin-3 motors by doublecortin family proteins. Mol Cell. 47:707-721.

Monroy, B.Y., T.C. Tan, J.M. Oclaman, J.S. Han, S. Simo, S. Niwa, D.W. Nowakowski, R.J. McKenney, and K.M. Ori-McKenney. 2020. A Combinatorial MAP Code Dictates Polarized Microtubule Transport. Dev Cell. 53:60-72 e64.

Moores, C.A., M. Perderiset, C. Kappeler, S. Kain, D. Drummond, S.J. Perkins, J. Chelly, R. Cross, A. Houdusse, and F. Francis. 2006. Distinct roles of doublecortin modulating the microtubule cytoskeleton. EMBO J. 25:4448-4457.

Rafiei, A., S. Cruz Tetlalmatzi, C.H. Edrington, L. Lee, D.A. Crowder, D.J. Saltzberg, A. Sali, G. Brouhard, and D.C. Schriemer. 2022. Doublecortin engages the microtubule lattice through a cooperative binding mode involving its C-terminal domain. eLife. 11.

Sun, T., Y. Li, T. Li, H. Ma, Y. Guo, X. Jiang, M. Hou, S. Huang, and Z. Chen. 2017. JIP1 and JIP3 cooperate to mediate TrkB anterograde axonal transport by activating kinesin-1. Cell Mol Life Sci. 74:4027-4044.

Zahavi, E.E., J.J.A. Hummel, Y. Han, C. Bar, R. Stucchi, M. Altelaar, and C.C. Hoogenraad. 2021. Combined kinesin-1 and kinesin-3 activity drives axonal trafficking of TrkB receptors in Rab6 carriers. Dev Cell. 56:1552-1554.